

# Impact of dust addition on the microbial food web under present and future conditions of pH and temperature

Julie Dinasquet[1,2*], Estelle Bigeard[3], Frédéric Gazeau[4], Farooq Azam[1], Cécile Guieu[4], Emilio Marañón[5], Céline Ridame[6], France Van Wambeke[7], Ingrid Obernosterer[2] and Anne-Claire Baudoux[3]

[1] Marine Biology Research Division, Scripps Institution of Oceanography, UCSD, USA

[2] Sorbonne Université, CNRS, Laboratoire d'Océanographie Microbienne, LOMIC, France

[3] Sorbonne Université, CNRS, Station Biologique de Roscoff, UMR 7144 Adaptation et Diversité en Milieu Marin, France

[4] Sorbonne Université, CNRS, Laboratoire d'Océanographie de Villefranche, LOV, 06230 Villefranche-sur-Mer, France

[5] Department of Ecology and Animal Biology, Universidade de Vigo, Spain

[6] CNRS-INSU/IRD/MNHN/UPMC, LOCEAN: Laboratoire d'Océanographie et du Climat: Expérimentation et Approches Numériques, UMR 7159

[7] Aix-Marseille Université, CNRS/INSU, Université de Toulon, IRD, Mediterranean Institute of Oceanography, UM110, France

*Corresponding: jdinasquet@ucsd.edu, present address: Center for Aerosol Impact on Chemistry of the Environment (CAICE), Scripps Institution of Oceanography, UCSD, USA

**Keywords:** bacteria, microeukaryotes, virus, community composition, top-down





## Abstract

In the oligotrophic waters of the Mediterranean Sea, during the stratification period, the microbial loop relies on pulsed inputs of nutrients through atmospheric deposition of aerosols from both natural (Saharan dust) and anthropogenic origins. While the influence of dust deposition on microbial processes and community composition is still not fully constrained, the extent to which future environmental conditions will affect dust inputs and the microbial response is not known. The impact of atmospheric wet dust deposition was studied both under present and future (warming and acidification) environmental conditions through experiments in 300 L climate reactors. Three dust addition experiments were performed with surface seawater collected from the Tyrrhenian Sea, Ionian Sea and Algerian basin in the Western Mediterranean Sea during the PEACETIME cruise in May-June 2017. Top-down controls on bacteria, viral processes and community, as well as microbial community structure (16S and 18S rDNA amplicon sequencing) were followed over the 3-4 days experiments. Different microbial and viral responses to dust were observed rapidly after addition and were most of the time higher when combined to future environmental conditions. The input of nutrients and trace metals changed the microbial ecosystem from bottom-up limited to a top-down controlled bacterial community, likely from grazing and induced lysogeny. The composition of mixotrophic microeukaryotes and phototrophic prokaryotes was also altered. Overall, these results suggest that the effect of dust deposition on the microbial loop is dependent on the initial microbial assemblage and metabolic state of the tested water, and that predicted warming, and acidification will intensify these responses, affecting food web processes and biogeochemical cycles.



## 1. Introduction

Input of essential nutrients and trace metals through aerosol deposition is crucial to the ocean
surface water biogeochemistry and productivity (at the global scale: *e.g.,* Mahowald et al., 2017;
in the Mediterranean Sea: *e.g.,* Guieu and Ridame, 2020) with episodic fertilization events
driving microbial processes in oligotrophic regions such as the Pacific Ocean, the Southern
Ocean and the Mediterranean Sea.
The summer Mediterranean food web is characterized by low primary production (PP) and
heterotrophic prokaryotic production (more classically abbreviated as BP for bacterial
production) constrained by nutrient availability further limiting dissolved organic matter (DOM)
utilization and export, resulting in DOM accumulation. Therefore, inputs of bioavailable
nutrients through deposition of atmospheric particles are essential to this microbial ecosystem.
Indeed, these nutrient pulses have been shown to support microbial processes but the degree to
which the microbial food web is affected might be dependent on the degree of oligotrophy of the
water (Marín-Beltrán et al., 2019; Marañon et al., 2010).
In the Mediterranean Sea, dust deposition stimulates PP and $N_2$ fixation (Guieu et al., 2014;
Ridame et al., 2011) but also BP, bacterial respiration, virus production, grazing activities, and
can alter the composition of the microbial community (*e.g.,* Pulido-Villena et al., 2014; Tsiola et
al., 2017; Guo et al., 2016; Pitta et al., 2017; Marín-Beltrán et al., 2019). Overall, in such
oligotrophic system, dust deposition appears to predominantly promote heterotrophic activity
which will increase respiration rates and $CO_2$ release.
Anthropogenic $CO_2$ emissions are projected to induce an increase in seawater temperature
and an accumulation of $CO_2$ in the ocean, leading to its acidification and an alteration of ocean
carbonate chemistry (IPCC, 2014). In response to ocean warming and increased stratification,



low nutrient low chlorophyll (LNLC) regions such as the Mediterranean Sea, are projected to
expand in the future (Durrieu de Madron et al., 2011). Moreover, dust deposition is also expected
to increase due to desertification (Moulin and Chiapello, 2006). Hence, in the future ocean, the
microbial food web might become even more dependent on atmospheric deposition of nutrients.
Expected increased temperature and acidification might have complex effects on the microbial
loop by modifying microbial and viral and community (*e.g.,* Highfield et al., 2017; Krause et al.,
2012; Hu et al., 2021; Allen et al., 2020; Malits et al., 2021). While increasing temperature in
combination with nutrient input might enhance heterotrophic bacterial growth (Degerman et al.,
2012; Morán et al., 2020) more than PP (Marañón et al., 2018), future environmental conditions
could push even further this microbial community towards heterotrophy. But so far, the role of
dust on the microbial food web in future climate scenarios is unknown.
Here, we studied the response of Mediterranean microbial and viral communities (*i.e.,* viral
strategies, microbial growth, and controls, as well as community composition) to simulated wet
Saharan dust deposition during onboard minicosm experiments conducted in three different
basins of the Western and Central Mediterranean Sea under present and future projected
conditions of temperature and pH. To our knowledge, this is the first study assessing the effect of
atmospheric deposition on the microbial food web under future environmental conditions.



## 2. Material & Method

### *2.1 Experimental set-up*

During the ProcEss studies at the Air-sEa Interface after dust deposition in the

MEditerranean sea project cruise (PEACETIME), onboard the R/V "Pourquoi Pas ?" in

May/June 2017, three experiments were conducted in 300 L climate reactors (minicosms) filled

with surface seawater collected at three different stations (Table 1), in the Tyrrhenian Sea (TYR),

Ionian Sea (ION) and in the Algerian basin (FAST). The experimental set-up is described in

details in Gazeau et al. (2020). Briefly, the experiments were conducted for 3 days (TYR and

ION) and 4 days (FAST) in trace metal free conditions, under light, temperature and pH-

controlled conditions following ambient or future projected conditions of temperature and pH.

For each experiment, the biogeochemical evolution of the water, after dust deposition, under

present and future environmental conditions was followed in three duplicate treatments: i)

CONTROL (C1, C2) with no dust addition and under present pH and temperature conditions, ii)

DUST (D1, D2) with dust addition under present environmental conditions and iii)

GREENHOUSE (G1, G2) with dust addition under projected temperature and pH for 2100

(IPCC, 2014; ca. +3 °C and -0.3 pH units). The same dust analog was used as during the DUNE

2009 experiments described in Desboeufs et al. (2014) and the same dust wet flux of 10 g m$^{-2}$

was simulated. Such deposition event represents a high but realistic scenario, as several studies

reported even higher short deposition events in this area of the Mediterranean Sea (Ternon et al.,

2010; Bonnet and Guieu, 2006; Loÿe-Pilot and Martin, 1996). After mixing the dust analog (3.6

g) in 2 L of ultrahigh-purity water, this solution was sprayed at the surface of the dust amended

treatments (D1, D2 and G1, G2; Gazeau et al., 2020).



Samples were taken at t-12h (while filling the tanks), t0 (just before dust addition), t1h,
t6h, t12h, t24h, t48h, t72h and t96h (after dust addition, and t96h only for FAST).
*2.2. Growth rates, mortality, and top down controls*
BP was estimated at all sampling points from rates of $^3$H-Leucine incorporation
(Kirchman et al., 1985; Smith and Azam, 1992) as described in Gazeau et al. (2021). Briefly,
triplicate 1.5 mL samples and one blank were incubated in the dark for 1-2 h in two temperature-
controlled incubators maintained respectively at ambient temperature for C1, C2, D1 and D2 and
at ambient temperature +3 °C for G1 and G2. HB, *Synechococcus*, picoeukaryotes and
heterotrohic nanoflagellates (HNF) abundances were measured by flow cytometry as described
in Gazeau et al. (2020). Bacterial cell specific growth rates were estimated assuming exponential
growth and a carbon to cell ration of 20 fg C cell$^{-1}$ (Lee and Fuhrman, 1987). Net growth rates
(h$^{-1}$) were calculated from the exponential phase of growth of BP, abundances of *Synechococcus*
and picoeukaryotes cells, observable from at least three successive sampling points. Mortality
was estimated as the difference between HB present between two successive sampling points and
those produced during that time.
*2.3. Viral abundance, production and life strategy*
Virus abundances were determined on glutaraldehyde fixed samples (0.5% final
concentration, Grade II, Sigma Aldrich, St Louis, MO, USA) stored at -80 °C until analysis. Flow
cytometry analysis was performed as described by Brussaard (2004). Briefly, samples were
thawed at 37 °C, diluted in 0.2 µm filtered autoclaved TE buffer (10:1 Tris-EDTA, pH 8) and
stained with SYBR-Green I (0.5 × 10$^{-4}$ of the commercial stock, Life Technologies, Saint-Aubin,
France) for 10 min at 80 °C. Virus particles were discriminated based on their green fluorescence



and SSC during 1 min analyses (Fig. S1). All cytogram analyses were performed with the Flowing
Software freeware (Turku Center of Biotechnology, Finland).
Viral production and bacterial losses due to phages were assessed by the virus reduction approach
(Weinbauer et al., 2010) at t0, t24 h and t48h in all six minicosms. Briefly, 3 L of seawater were-
filtered through 1.2-µm-pore-size polycarbonate filter (Whatman©), and heterotrophic
prokaryotes (HB, filtrate) were concentrated by ultrafiltration (0.22 µm pore size, Vivaflow 200©
polyethersulfone, PES) down to a volume of 50 mL. Virus-free water was obtained by filtering 1
L of seawater through a 30 kDa pore-size cartridge (Vivaflow 200©, PES). Six mixtures of HB
concentrate (2 mL) diluted in virus-free water (23 mL) were prepared and distributed into 50 mL
Falcon tubes. Three of the tubes were incubated as controls, while the other three were inoculated
with mitomycin C (Sigma-Aldrich, 1 µg mL$^{-1}$ final concentration) as inducing agent of the lytic
cycle in lysogenic bacteria. All tubes were incubated in darkness in two temperature-controlled
incubators maintained respectively at ambient temperature for C1, C2, D1 and D2 and at ambient
temperature +3 °C for G1 and G2. Samples for HB and viral abundances were collected every 6 h
for a total incubation period of 18 h.
The estimation of virus-mediated mortality of HB was performed according to Weinbauer et al.
(2002) and Winter et al. (2004). Briefly, increase in virus abundance in the control tubes represents
lytic viral production (VPL), and an increase in mitomycin C treatments represents total (VPT),
*i.e.,* lytic plus lysogenic, viral production. The difference between VPT and VPL represents
lysogenic production (VPLG). The frequency of lytically infected cells (FLIC) and the frequency
of lysogenic cells (FLC) were calculated as:
$FLIC = 100 \times VPL / BS \times HB_i$                                             (1)
$FLC = 100 \times VPLG / BS \times HB_i$                                             (2)



where $HB_i$ is the initial HB abundance in the viral production experiment and BS is a theoretical
burst size of 20 viruses per infected cell (averaged BS in marine oligotrophic waters, Parada et al.,

2006).


*2.4 DNA sampling, sequencing and sequence analysis*
To study the temporal dynamics of the microbial diversity, water samples (3 L) were
collected in acid-washed containers from each minicosm at t0, t24h, and at the end of the
experiments (t72h at TYR and ION and t96h at FAST). Samples were filtered onto 0.2 µm PES
filters (Sterivex©) and stored at -80 °C until DNA extraction. Nucleic acids were extracted from
the filters using a phenol-chloroform method and DNA was then purified using filter columns from
NucleoSpin® PlantII kit (Macherey-Nagel©) following a modified protocol. DNA extracts were
quantified and normalized at 5ng µL$^{-1}$ and used as templates for PCR amplification of the V4
region of the 18S rRNA (~380 bp) using the primers TAReuk454FWD1 and TAReukREV3
(Stoeck et al., 2010) and the V4-V5 region of the 16S rRNA (~411 bp) using the primers 515F-Y
(5'-GTGYCAGCMGCCGCGGTAA) and 926R-R (5'-CCGYCAATTYMTTTRAGTTT) (Parada
et al., 2016). Following polymerase chain reactions, DNA amplicons were purified, quantified and
sent to Genotoul (https://www.genotoul.fr/, Toulouse, France) for high throughput sequencing
using paired-end 2x250bp Illumina MiSeq. Note that although we used universal primer, Archaea
were mostly not detected and the prokaryotic heterotrophic communities corresponded essentially
to Eubacteria, therefore the taxonomic description referred to the general term 'bacterial
communities'
All reads were processed using the Quantitative Insight Into Microbial Ecology 2 pipeline
(QIIME2 v2020.2, Bolyen et al., 2019). Reads were truncated 350bp based on sequencing



quality, denoised, merged and chimera-checked using DADA2 (Callahan et al., 2016). A total of
714 and 3070 amplicon sequence variants (ASVs) were obtained for 16S and 18S respectively.
Taxonomy assignments were made against the database SILVA 132 (Quast et al., 2013) for 16S
and PR2 (Guillou et al., 2013) for 18S. All sequences associated with this study have been
deposited under the BioProject ID: PRJNA693966.
*2.5 Statistics*

Alpha and beta-diversity indices for community composition were estimated after

randomized subsampling to 26000 reads for 16S rDNA and 19000 reads for 18S rDNA. Analysis
were run in QIIME 2 and in Primer v.6 software package (Clarke and Warwick, 2001).
Differences between the samples richness and diversity were assessed using Kruskal-Wallis
pairwise test. Beta diversity were run on Bray Curtis dissimilarity. Differences between samples'
beta diversity were tested using PERMANOVA (Permutational Multivariate Analysis of
Variance) with pairwise test and 999 permutations. The sequences contributing most to the
dissimilarity between clusters were identified using SIMPER (similarity percentage). A linear
mixed model was performed using the R software (R Core Team, 2020) using the nlme package
(Pinheiro et al., 2014) to test if the amended treatments differed from the controls at t24h and
t72h or t96h.



## 3. Results

### 3.1. Microbial growth, mortality and top-down controls

Significant increases in heterotrophic bacterial cell specific growth rates were observed in all experiments with dust under D and G (Fig. 1, $p \leq 0.016$ after 24 h and 72 h) relative to C, the highest growth rates relative to C were observed already 24 h after dust seeding. Bacterial net growth rates were also higher in D and especially in G relative to C (Table 2). *Synechococcus* and picoeukaryotes net growth rates showed a similar trend (Table 2). Heterotrophic bacterial mortality was also higher than in C especially at TYR and in G at ION and FAST (Fig. 1). Over the course of the three experiments, the slope of the linear regression between bacterial biomass and bacterial production was below 0.4 in the three treatments suggesting a weak bottom up control (Fig. 2A; Ducklow, 1992). The slope decreased in D and G relative to C. Overall, the top down index, as described by Morán et al. (2017), was higher in G (0.92) relative to C and D (0.80). The relationship between log transformed HNF and bacterial abundance (Fig. 3B), plotted according to the model in Gasol (1994), showed that HNF were below the MRA (Mean realized HNF abundance) in all treatments, suggesting a top down control of HNF abundance. HNF and bacteria were weakly coupled in all treatments. The relationship between total viruses and bacterial abundance was weaker in D and G relative to C (Fig. S2).

### 3.2. Viral dynamics and processes

The abundance and production of virus-like particles (VLP) increased following an east to west gradient (Table 1). Viral strategy (lysogenic vs. lytic replication) was also different between stations, with a higher frequency of lysogenic cells (FLC) at TYR and ION (23 and


19%, respectively Table 1) and a higher frequency of lytically infected cells (FLIC) at FAST
(43%, Table 1).

During TYR and ION experiments, the relative contribution of VLP populations was similar

and stable over time with Low DNA viruses representing over 80% of the community (Figs. 3
and S3). The Low DNA VLP abundance was however slightly higher in D and G relative to C
after 24 h at TYR and significantly higher at ION after 48h (p = 0.037; Fig. S3).  In contrast to
the other two stations, at FAST, Giruses were also present and increased in all treatments but
especially in G where they made up to 9% of the viral community at the end of the experiment
(Figs. 3 and S3). The abundance of high DNA viruses at FAST also increased independent of
treatments and accounted for 16 – 18% of the community at the end of the experiment (Figs. 3
and S3).

The sampling strategy for production and life strategies of HB viruses allowed to

discriminate independently the effect of i) greenhouse conditions (sampling at T0 before dust
addition), ii) dust addition (sampling at T24) and the combined effects of dust addition and
greenhouse. Lytic viral production (VPL) increased significantly at T0 in G at TYR and ION
compared to C (p ≤ 0.036). The addition of dust induced higher VPL in D at TYR compared to C
(Fig.1). No significant impact of dust on VPL was observed in G compared to D after 24h for
any of the experiments. Changes in viral infection strategy were observed with G conditions at
T0 where, FLC decreased relative to the non-G treatments at TYR and ION, and especially at
FAST (p = 0.047). FLIC increased slightly in G at TYR and ION already at T0. Dust addition
had no detectable significant effect on this parameter for any experiments. Looking at the
relative share between lytic and lysogenic infection, dust addition favored lytic infection at TYR
(no lysogenic bacteria were observed after 24h) but the contribution of both infection strategies



remained unchanged compared to C at ION and FAST. Greenhouse conditions also favored
replication through lytic cycle already at T0 for all three experiments and this trend was not
impacted by dust addition.
*3.3. Microbial community composition*

Microbial community structure, bacteria and micro-eukaryotes from 16S rDNA and 18S

rDNA sequencing respectively, responded to dust addition in all three experiments relative to C
(Figs. 4 and 5). After quality controls, reads were assigned to 714 and 1443 ASVs for 16S and
18S respectively.
*3.3.1. Bacterial community composition*

The initial community composition (t-12h) was significantly different at the three stations

(PERMANOVA; p = 0.001, Fig. S4a, S5). Rapid and significant changes in the bacterial
community composition were observed already 24 h after dust addition (Fig. 4). Despite the
initial different communities, the three stations appeared to converge towards a closer
community composition in response to dust addition (Fig. S5). At TYR, communities in D and G
significantly changed 24 h after dust addition (PERMANOVA; p = 0.001). This cluster presented
no significant differences between treatments (D and G) or time (24 and 72 h). The differences
between C and D/G were attributed to a relative increase of ASVs related to different
*Alteromonas* sp., OM60 and *Pseudophaeobacter* sp. and *Erythrobacter* sp.; contribution of
ASVs related to SAR11 and Verrucomicrobia and *Synechococcus* decreased (Table S1a). At
ION, the bacterial community composition significantly changed 24 h after dust addition
(PERMANOVA; p = 0.001) and was significantly different between D and G (PERMANOVA; p
= 0.032). As observed at TYR, no further change occurred between 24 h and the end of the





experiment (72 h; Fig. 4). The difference between the controls and dust amended minicosms
were assigned to an increase of ASVs related to different *Alteromonas* sp., *Erythrobacter* sp.,
*Dokdonia* sp. and OM60, and a decrease of ASVs related to SAR11, *Synechococcus*,
Verrucomicrobia, Rhodospirillales and some Flavobacteria (Table S1b). Several ASVs related to
*Alteromonas* sp., *Synechococcus* sp. and *Erythrobacter* sp. were further enriched in G compared
D while *Dokdonia* sp. was mainly present in D. At FAST, the bacterial community after 24 h
only significantly changed in G (PERMANOVA; $p = 0.011$; Fig. 4). However, after 96 h, the
community in D and G were similar and appeared to transition back to the initial state at 96 h
(PERMANOVA; $p = 0.077$). The higher relative abundance in *Erythrobacter* sp., *Synechoccocus*
sp., different ASVs related to *Alteromonas* sp. and Flavobacteria appeared to contribute mainly
to the difference between C and D/G (Table S1) while ASVs related to SAR11,
Verrucomicrobia, *Celeribacter* sp. *Thalassobius* sp. and Rhodospirillales were mainly present in
C (Table S1c).
*3.3.2 Nano- and micro-eukaryotes community composition*

The diversity of initial community was large (Fig. S5) and significantly different at the three

stations (PERMANOVA; $p = 0.001$; Fig. S4b). At TYR, the nano- and micro-eukaryotes
community responded rapidly (24 h) to dust addition (PERMANOVA; $p = 0.003$). This initial
high diversity disappeared after 72 h, with similar communities in all minicosms (Fig. S5). They
were significantly different from initial and t24h communities ($p = 0.002$ and $0.03$ respectively;
Fig 5) in D/G. The variations at t24h were attributed to changes in the dinoflagellate
communities in particular to an increase in ASVs related to *Heterocapsa rotundata*,
Gymnodiniales and Gonyaulacales as well as to an increase in Chlorophyta (Table S2a). At ION,
no significant changes were observed between C and D/G after 24 h. However, after 72 h, the





communities were significantly different in D (p = 0.018) and G (p = 0.05) compared to the
communities at t24h in these treatments (Table S2B). In D, diversity was significantly higher at
t72h compared to t24h and to C at the same sampling time (p = 0.036). In contrast, diversity in G
at t72h was lower than at t24h and lower to the one observed in C at the same sampling time (p =
0.066; Fig S6). These differences were mainly attributed to changes in ASVs related to
dinoflagellates and to the increase at t72h of *Emiliana huxleyi* and Chlorophyta in D and G,
respectively (Table S2b). At FAST, significant differences were observed between the controls
and initial communities compared to the dust amended (D and G) treatments at t24h (p = 0.036).
No major differences were observed between D/G at t24h and t96h (p = 0.06). The differences
were mainly attributed to changes in dinoflagellates ASVs and to an increase in Acantharea and
*Emiliana huxleyi* in D and G treatments at t96h (Table S2c).



## 4. Discussion


Pulsed inputs of essential nutrients and trace metals through aerosol deposition are crucial to
surface microbial communities in LNLC regions such as the Mediterranean Sea (reviewed in
Guieu and Ridame, 2020). Here we assessed the impact of dust deposition on the late spring
microbial loop under present and future environmental conditions on the surface water of three
different Mediterranean basins (Tyrrhenian, TYR; Ionian, ION; and Algerian, FAST). The initial
conditions at the three sampled stations for the onboard experiments are described in more
details in Gazeau et al. (2020). Briefly, very low levels of dissolved inorganic nutrients were
measured at all three stations, highlighting the oligotrophic status of the waters, typical of the
stratified conditions observed in the Mediterranean Sea in late spring/early summer (*e.g.,* Bosc et
al., 2004; D'Ortenzio et al., 2005). Despite similar total chl. *a* concentrations at the three stations
(Gazeau et al., 2020), PP was higher at FAST (Table 1, Gazeau et al., 2021; Marañón et al.,
2021). The initial microbial communities differed substantially between the three stations as
shown by pigments (Gazeau et al., 2020), 18S and 16S rDNA sequencing (this study). DOC
concentrations were slightly higher at TYR where PP was the lowest (Gazeau et al., 2021). HB,
HNF abundances (Gazeau et al., 2020), as well as viral abundance and production increased
following the east to west gradient of the initial water conditions.
The dust addition induced similar nitrate + nitrite ($NO_x$) and dissolved inorganic phosphate
(DIP) release during all three experiments. Rapid changes were observed on plankton stocks and
metabolisms, suggesting that the impact of dust deposition is constrained by the initial
composition and metabolic state of the investigated community (Gazeau et al., 2020; 2021).
While no direct effect of warming and acidification was observed on the amount of nutrient
released from dust, Gazeau et al., (2020, 2021) showed that biological processes were generally



enhanced by these conditions and suggested that deposition may weaken the biological pump in
future climate conditions. Here we are further investigating how dust addition in present and
future conditions affected, on a short-term scale (≤ 4 days), the microbial trophic interactions and
community composition.
*4.1. Trophic interactions after dust addition under present and future conditions*

Parallel nutrient enrichment incubations conducted in darkness showed that *in situ*

heterotrophic bacterioplankton communities, were N, P co-limited at TYR, mainly P limited at
ION and N limited at FAST (Van Wambeke et al., 2020). However, the HB appeared to be
weakly bottom up controlled (Ducklow, 1992) in our experiment especially in D and G (Fig 2a).
Such top-down control on the bacterioplankton has been previously observed in the
Mediterranean Sea (Siokou-Frangou et al., 2010) and might increase under future conditions as
suggested by the higher top-down index in G (G = 0.92 vs. C/D= 0.80, Morán et al., 2017).

Bacterial mortality increased relative to controls in D and G at TYR, and only in G at ION

and FAST. The weak coupling between bacteria and viruses, as well as the increased virus
production and relative abundance of lytic cells (see below), only explained a small fraction of
the estimated bacterial mortality (max. 17%), suggesting an additional grazing pressure on
bacteria. HNF abundances increased in D at TYR and at all stations in G (Gazeau et al., 2020),
which could explain the increased bacterial mortality. Increased grazing rate by HNF on bacteria
with dust addition has been previously reported in the Eastern Mediterranean Sea (Tsiola et al.,
2017). While our results suggest a strong grazing pressure on bacteria, HNF appeared to be top-
down controlled as well (Gasol, 1994, Fig 3b), potentially by the increasing populations of
mixotrophic dinoflagellates or Giruses (see below). It is also possible that HB were grazed by



mixotrophic nanoflagellates or by larger protozoans, or that the HNF abundance was
underestimated by flow cytometry.

Considering the seasonal impact of grazing and viral mortality in the Mediterranean Sea,

where higher grazing pressure and lysogeny were observed in the stratified nutrient-limited
waters in summer (Sánchez et al., 2020), it will be important to further study the seasonal impact
of dust deposition on trophic interactions and indirect cascading impact on microbial dynamics
and community composition.

*4.2. Viral processes and community during dust enrichment in present and future conditions*

Viruses represent pivotal components of the marine food web, influencing genome evolution,

community dynamics, and ecosystem biogeochemistry (Suttle, 2007). The environmental and
evolutionary implications of viral infection differ depending on whether viruses establish lytic or
lysogenic infections. Lytic infections produce virion progeny and result in cell destruction while
viruses undergoing lysogenic infections can replicate as "dormant" prophages without producing
virions or can switch to a lytic productive cycle upon an induction event. Understanding how
viral processes are impacted by changes in environmental conditions, is thus crucial to better
constrain microbial mortality and cascading impacts on marine ecosystems. Aerosol deposition
was already identified as a factor that stimulates virus production and viral induced mortality of
bacteria in the Mediterranean Sea (Pulido-Villena et al., 2014; Tsiola et al., 2017) while the
impact of future environmental conditions remains more controversial ( Larsen et al., 2008;
Brussaard et al., 2013; Maat et al., 2014; Vaqué et al., 2019; Malits et al., 2021). The combined



effect of aerosol deposition and future conditions of temperature and pH on the viral
compartment has, to our knowledge, never been investigated.
The rapid changes in viral production and lifestyle observed in all three experiments support the
idea that the viral component is sensitive to the environmental variability even on short (hourly)-
time scales. The dynamics in viral activities was however impacted differently depending on the
treatments and the experiments. Viral production increased in D and G at TYR and only in G at
ION and FAST. Regarding the G treatments, increase in viral production was detected before
dust addition for all three experiments and remained mostly unchanged for the remaining of the
incubation. This suggests that water warming, and acidification were responsible for most
changes in viral activities while dusts had no detectable impact in such conditions regardless of
the studied station. Based on our results, the most likely explanation for observed changes in
viral production is an activation of a lysogenic to lytic switch. The factors that result in prophage
induction are still not well constrained, but nutrients pulses and elevated temperatures have been
identified as potential stressors (Danovaro et al., 2011 and references therein). Consistent with
the observation of N, P co-limited bacterial community at TYR, it is likely that nutrients released
from dust upon deposition to surface water activate the productive cycle of temperate viruses at
this station. Such mechanism was also speculated during another dust addition study (Pulido-
Villena et al., 2014). Under future conditions (G), the low proportion of lysogens was associated
to higher frequency of lytically infected cells relative to C and D at TYR and ION. These trends
probably reflect an indirect effect of enhanced bacterial growth with increased temperature not
only on prophage induction (Danovaro et al., 2011; Vaqué et al., 2019; Mojica and Brussaard,
2014) but also on the kinetics of lytic infections. Intriguingly, the enhanced viral production did
not translate into marked changes in viral abundance. The abundance of Low DNA virus





population, which typically comprises virus of bacteria, actually decreased between t0 and t48h
pointing to possible viral decay, potentially related to an adsorption onto dust particles
(Weinbauer et al., 2009;Yamada et al., 2020) and the potential export of viral particle to deeper
water layers (Van Wambeke et al. 2020). While recurrent patterns emerged from this study, the
amplitude of viral responses varied between the experiments. At TYR, where heterotrophic
metabolism was higher, the dust addition induced higher viral production relative to controls
than at the two other sites, which suggests that viral processes, as other microbial processes, are
dependent on the initial metabolic status of the water.
Overall, no marked changes were observed for viral communities and abundances after dust
addition, both under present and future conditions relative to controls, except at FAST where the
abundance of Girus population increased significantly in G from t24h until the end of the
experiment. Giruses typically comprise large double stranded DNA viruses that infect
nanoeukaryotes including photosynthetic (microalgae) and heterotrophic (HNF, amoeba,
choanoflagellate) organisms (Brussaard and Martinez, 2008; Needham et al., 2019; Fischer et al.,
2010; Martínez et al., 2014). The presence of Giruses at FAST in this treatment might be
explained by the increase in nano-eukaryote abundances at t72h and their decline after 96 h of
incubation (Gazeau et al., 2020). The coccolithophore *Emiliania huxleyi* appears as one of the
potential host candidates for these Giruses. The abundance of *E. huxleyi* increased in D and G at
this station and this phytoplankter is known to be infected by such giant viruses (Jacquet et al.,
2002; Schroeder et al., 2002; Pagarete et al., 2011). It is not clear from our results whether
increased Girus abundance is due to the greenhouse effect only (as discussed above for viruses of
HB) or the combination of dust addition and greenhouse effects. While temperature warming
was shown to accelerate viral production in several virus – phytoplankton systems (Mojica and

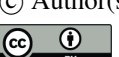



Brussaard 2014, Demory et al. 2017), a temperature-induced resistance to viral infection was
specifically observed in *E. huxleyi* (Kendrick et al., 2014).  Previous experiments have also
reported a negative impact of acidification on *E. huxleyi* virus dynamics (Larsen et al., 2008). By
contrast, nutrient release following dust seeding could indirectly stimulate *E. huxleyi* virus
production (Bratbak et al., 1993) or induced switching between non-lethal temperate to lethal
lytic stage (Knowles et al.,2020) under future conditions.  Targeted analyses are of course
required to identify the viral populations selected in G and the outcomes of their infection.
Nonetheless, this is the first time, to our knowledge, that dust deposition and enhanced
temperature and acidification have been shown to induce the proliferation of Giruses. The impact
of dust deposition under future environmental conditions on the viral infections processes could
have significant consequences for microbial evolution, food web processes, biogeochemical
cycles, and carbon sequestration.

*4.3 Microbial community dynamic after dust addition under present and future conditions*
While changes in bacterial community composition during various type of dust addition
experiments have shown only minor transient responses (*e.g.,* Marañon et al., 2010; Hill et al.,
2010; Laghdass et al., 2011; Pulido-Villena et al., 2014; Marín-Beltrán et al., 2019), here
microbial community structure showed quick, significant and sustained changes in response to
dust addition in all three experiments. Similar to other parameters observed during these
experiments (discussed above and in Gazeau et al., 2020; Gazeau et al., 2021), the degree of
response in terms of community composition was specific to the tested waters.





At TYR, where primary production was low, only transient changes after 24 h of incubation
were observed, before the micro-eukaryotes community converged back close to initial
conditions. In contrast, the bacterial community significantly and rapidly changed after 24 h and
remained different after 72 h. At FAST, where the addition of dust appeared to promote
autotrophic processes, the micro-eukaryotes community responded quickly 24 h after dust
addition, while minor and delayed changes, probably related to the lower BP growth rates
compared to the other tested waters, were observed in the bacterial community. At ION both
eukaryotes and bacterial community responded to dust addition. The delayed response of micro-
eukaryotes after 72 h compared to the quick bacterial response at 24 h suggests that HB were
better at competing for nutrient inputs at this station and that autotrophic processes may be
responding to bacterial nutrient regeneration after a lag phase, further suggesting the tight
coupling between heterotrophic bacteria and phytoplankton at this station. The combined effect
of decreased pH and elevated temperature on marine microbes is not yet well understood
(reviewed in O'Brien et al., 2016). The absence of significant community changes at TYR and
FAST while changes were observed at ION, suggests that the response might be dependent on
other environmental factors, which need to be further studied.
Dust addition likely favors certain group of micro-organisms, suggesting a quicker response
of fast growing/copiotrophic groups as well as the increase of specialized functional groups (Guo
et al., 2016; Westrich et al., 2016; Maki et al., 2016). Potential toxicity effects of metals released
from dust/aerosols on certain micro-organisms have also been reported (Paytan et al., 2009;
Rahav et al., 2020). Here, the micro-eukaryotic community was dominated by a diverse group of
dinoflagellates which were responsible for the main variations between treatments at all stations.
The overwhelming abundance of dinoflagellates sequences over other micro-eukaryotes could be



biased by the large genomes and multiple ribosomal gene copies per genome found in
dinoflagellates (Zhu et al., 2005) or due to their preferential amplification. However, the
dominance of dinoflagellates in surface water at this time of the year in the Mediterranean Sea is
not uncommon (García-Gómez et al., 2020) and was also observed in surface waters of the three
sampled stations by Imaging Flow Cytobot (Marañón et al., 2021). While pigment data suggest
an increase of haptophytes and pelagophytes in D (Gazeau et al., 2020), the sequencing data only
show the presence of *Emiliana huxleyi* as responsible for some of the community changes after
dust addition at ION and FAST. These pigments could also indicate the presence of
dinoflagellates through tertiary endosymbiosis, in particular *Karlodinium* sp. (Yoon et al., 2002;
Zapata et al., 2012), which is an important mixotrophic dinoflagellate (Calbet et al., 2011)
observed in D and G at ION and FAST. The variations in dinoflagellate groups might have
important trophic impacts due to their diverse mixotrophic states (Stoecker et al., 2017) and the
effect of dust addition on mixotrophic interactions should be further studied to better understand
the cascading impact of dust on food webs and the biological pump.
Positive to toxic impacts on cyanobacteria have been reported from atmospheric deposition
experiments (*e.g.,* Paytan et al., 2009; Zhou et al., 2021). Here, *Synechococcus* appeared to be
inhibited at TYR while it was enhanced at ION and FAST, especially under future conditions
(this study, Gazeau et al., 2020). The same ASVs appeared to be inhibited at TYR and ION
while promoted at FAST and a different ASVs increased at ION. *Synechococcus* has recently
been shown to be stimulated by wet aerosol addition in P-limited conditions but inhibited in N-
limited conditions, in the South China Sea (Zhou et al., 2021). It was also shown to be repressed
by dust addition in nutrient limited tropical Atlantic (Marañon et al., 2010). This suggests that



different *Synechococcus* ecotypes (Sohm et al., 2016) might respond differently to dust addition
depending on the initial biogeochemical conditions of the water.

In the three experiments, the main bacterial ASVs responsible for the differences between

the control and treatments were closely related to different *Alteromonas* strains. *Alteromonas* are
ubiquitous in marine environment and can respond rapidly to nutrient pulses (López-Pérez and
Rodriguez-Valera, 2014). Some *Alteromonas* are capable to grow on a wide range of carbon
compounds (Pedler et al., 2014). They can produce iron binding ligands (Hogle et al., 2016) to
rapidly assimilate Fe released from dust. Thus, they could have significant consequences for the
marine carbon and Fe cycles during dust deposition events. Other copiotrophic γ-Proteobacteria,
such as *Vibrio,* have been observed to bloom after dust deposition in the Atlantic Ocean
(Westrich et al., 2016). Guo et al. (2016) using RNA sequencing, also show that γ-Proteobacteria
quickly outcompete α-Proteobacteria (mainly SAR11 and Rhodobacterales) that were initially
more active. Here, while SAR11 relative abundance decreased in all experiments after 24h, other
α-Proteobacteria related to the aerobic anoxygenic phototroph (AAP) *Erythrobacter* sp.,
increased in response to dust, in particular under future conditions. Other AAP, such as OM60,
also responded to dust addition in our experiment and in the Eastern Mediterranean Sea (Guo et
al., 2016). Fast growing AAP might quickly outcompete other HB by supplementing their
growth with light derived energy (*e.g.,* Koblížek, 2015). They have also been shown to be
stimulated by higher temperature (Sato-Takabe et al., 2019). AAP response to dust and future
conditions could have a significant role in marine biogeochemical cycles.
**5. Conclusion**

The microbial food web response to dust addition was dependent on the initial state of the

microbial community in the tested waters. A different response in trophic interactions and



community composition of the microbial food web, to the wet dust addition, was observed at
each station. Generally greater changes were observed in future conditions. Pulsed input of
nutrients and trace metals changed the microbial ecosystem from bottom-up limited to a top-
down controlled bacterial community, likely from grazing and induced lysogeny. The
composition of mixotrophic microeukaryotes and phototrophic prokaryotes was also altered.
Overall, the impact of such simulated pulsed nutrient deposition will depend on the initial
biogeochemical conditions of the ecosystem, with likely possible large impact on microbial
trophic interactions and community structure. All effects might be generally enhanced in future
climate scenarios. The impact of dust deposition on metabolic processes and consequences for
the carbon and nitrogen cycles and the biological pump based on these minicosm experiments
are further discussed in Gazeau et al. (2021), and the *in situ* effect of a wet dust deposition event
is explored in Van Wambeke et al. (2020), in this special issue.
**6. Data availability**
Guieu et al., Biogeochemical dataset collected during the PEACETIME cruise. SEANOE.
https://doi.org/10.17882/75747 (2020).All sequences associated with this study have been
deposited under the BioProject ID: PRJNA693966.
**7. Author contributions**
FG and CG designed the experiment. All authors participated in sampling or sample
processes. JD analyzed the data and wrote the paper with contributions from all authors.
**8. Competing interests**
The authors declare that they have no conflict of interest.



9. **Special issue statement**

This article is part of the special issue 'Atmospheric deposition in the low-nutrient–low-chlorophyll (LNLC) ocean: effects on marine life today and in the future (ACP/BG inter-journal SI)'. It is not associated with a conference.

10. **Financial support**

Part of this research was funded by the ANR CALYPSO attributed to ACB (ANR-15-CE01-0009). EM was supported by the Spanish Ministry of Science, Innovation and Universities through grant PGC2018-094553B-I00. JD was funded by a Marie Curie Actions-International Outgoing Fellowship (PIOF-GA-2013-629378).

11. **Acknowledgements**

This study is a contribution to the PEACETIME project (http://peacetime-project.org, https://doi.org/10.17600/17000300), a joint initiative of the MERMEX and ChArMEx components supported by CNRS-INSU, IFREMER, CEA, and Météo-France as part of the programme MISTRALS coordinated by INSU. PEACETIME was endorsed as a process study by GEOTRACES and is also a contribution to SOLAS. We gratefully acknowledge the onboard support from the captain and crew of the RV Pourquoi Pas? and of our chief scientists C. Guieu and K. Desboeufs. We also thank K. Djaoudi for her assistance in sampling the minicosms, P. Catala, B. Marie and M. Perez-Lorenzo with their assistance in measuring microbial abundance, DOC concentration and primary production.

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





## Tables and Figures

**Table1**: Initial conditions (t-12h) at the three stations sampled for the dust addition experiments. Other parameters are presented in more details in Gazeau et al. (2020; 2021)

| Variables | TYR | ION | FAST |
|---|---|---|---|
| Location | Tyrrhenian Basin | Ionian Basin | Algerian Basin |
| Coordinates | 39.34N, 12.60E | 35.49N, 19.78E | 37.95N,2.90E |
| Temperatures (°C) | 20.6 | 21.2 | 21.5 |
| DOC ($\mu$M)[2] | 72.2 | 70.2 | 69.6 |
| Chlorophyll $a$ ($\mu$g L$^{-1}$)[1] | 0.063 | 0.066 | 0.072 |
| BP (ng C L$^{-1}$ h$^{-1}$)[2] | 11.6 | 15.1 | 34.6 |
| Bacterial abundance (x10$^5$ cells mL$^{-1}$)[1] | 4.79 | 2.14 | 6.15 |
| Viral abundance (x 10$^6$ VLP mL$^{-1}$) | 3.01 | 1.44 | 2.79 |
| % Lysogenic bacteria FLC | 22.7 | 19.4 | 7.8 |
| % Lytic bacteria FLIC | 17.5 | 37.2 | 42.7 |
| Viral production (x 10$^4$ VLP mL$^{-1}$ h$^{-1}$) | 2.05 | 1.36 | 7.99 |
| HNF abundance (cells mL$^{-1}$)[1] | 110 | 53 | 126 |
| Diatoms (cells L$^{-1}$)[1] | 340 | 900 | 1460 |
| Dinoflagellates (cells L$^{-1}$)[1] | 2770 | 3000 | 3410 |
| Ciliates (cells L$^{-1}$)[1] | 270 | 380 | 770 |

DOC: dissolved organic carbon, * BP: heterotrophic prokaryotic production, HNF: Heterotrophic nanoflagellates
[1]Results presented in Gazeau et al. 2020
[2]Results presented in Gazeau et al. *2021*





**Table 2.** Net growth rates (h⁻¹) calculated from the exponential phase of growth of BP,
abundances of Synechococcus and picoeukaryotes cells, observable from at least three
successive sampling points. Values ± standard error are shown, as well as the period of
exponential phase (period, in days). nd: no significant exponential phase noted.

| | | | C1 | C2 | D1 | D2 | G1 | G2 |
|---|---|---|---|---|---|---|---|---|
| TYR | μapp BP | mean ± sd | 0.076 ± 0.025 | 0.066 ± 0.018 | 0.116 ± 0.008 | 0.194 ± 0.02 | 0.164 ± 0.019 | 0.1503 ± 0.003 |
| | | period | 0 - 0.5 | 0 - 0.5 | 0 - 0.5 | 0 - 0.5 | 0 - 0.5 | 0 - 0.5 |
| TYR | μapp syn | mean ± sd | nd | nd | nd | nd | 0.014 ± 0.05 | 0.033 ± 0.003 |
| | | Period | | | | | 2 - 3 | 2 - 3 |
| TYR | μapp picoeuk | mean ± sd | nd | nd | nd | nd | 0.024 ± 0.004 | nd |
| | | period | | | | | 2 - 3 | |
| ION | μapp BP | mean ± sd | 0.042 ± 0.007 | 0.041 ± 0.005 | 0.09 ± 0.02 | 0.14 ± 0.006 | 0.13 ± 0.01 | 0.14 ± 0.03 |
| | | Period | 0 - 0.5 | 0 - 0.5 | 0 - 0.5 | 0 - 0.5 | 0 - 0.5 | 0 - 0.5 |
| ION | μapp syn | mean ± sd | nd | nd | 0.011± 0.001 | 0.015 ±0.001 | 0.038± 0.002 | 0.045 ± 0.008 |
| | | Period | | | 0.5 - 2 | 0.5 - 2 | 0.5 - 2 | 0.5 - 2 |
| ION | μapp picoeuk | mean ± sd | 0.018 ± 0.001 | 0.012 ± 0.007 | 0.043 ± 0.014 | 0.034 ± 0.014 | 0.057 ± 0.012 | 0.053 ± 0.008 |
| | | Period | 0.5 - 3 | 0.5 - 2 | 0.5 - 2 | 0.5 - 2 | 0.5 - 2 | 0.5 - 2 |
| FAST | μapp BP | mean ± sd | 0.020 ± 0.002 | 0.026± 0.003 | 0.089 ± 0.014 | 0.090 ± 0.007 | 0.12 ± 0.005 | 0.16 ± 0.014 |
| | | Period | 0 - 0.5 | 0 - 0.5 | 0 - 0.5 | 0 - 0.5 | 0 - 0.5 | 0 - 0.5 |
| FAST | μapp syn | mean ± sd | 0.022 ± 0.002 | 0.024 ± 0.002 | 0.039 ± 0.001 | 0.045 ± 0.003 | 0.064 ± 0.001 | 0.063 ± 0.001 |
| | | Period | 0.5 - 2 | 0.5 - 2 | 0.5 - 2 | 0.5 - 2 | 0.5 - 2 | 0.5 - 2 |
| FST | μapp picoeuk | mean ± sd | 0.020 ± 0.002 | 0.012 ± 0.001 | 0.023± 0.004 | 0.026 ± 0.001 | 0.040 ± 0.002 | 0.034 ± 0.005 |
| | | Period | 0.5 - 2 | 0.5 - 2 | 0.5 - 2 | 0.5 - 2 | 0.5 - 2 | 0.5 - 2 |












**Figure legends:**
**Figure 1.** Bacterial and viral parameters in the three experiments (TYR, ION and FAST) in each minicosm
(D1, D2, G1 and G2). The values are normalized to the controls: the data are presented as the difference
between the treatments and the mean value of the duplicate controls. The first raw represents the
bacterial cell specific growth rates and relative mortality rates at t24h after dust addition. The second
raw represents the relative viral productions at t24h and at T0 for the G treatments. The last raw
represents the viral strategies: the percentages of lytic (FLIC) or lysogenic (FLC) cells at t24h and at T0 for
the G treatments.
**Figure 2.** (A) Log-log linear regression between bacterial biomass and bacterial production, dotted lines
represent linear regressions for each treatment. (B) Relationships between log HNF abundance and log
bacterial prey abundance. Solid black and dotted black lines corresponds to the Mean Realized HNF
Abundance (MRA) and theoretical Maximum Attainable HNF Abundance line (MAA) respectively. The
samples are grouped per treatments.
**Figure 3.** Relative abundance of viral populations at the initial (*in situ*: at t-12h before dust addition) and
final time points in all minicosms (C1, C2, D1, D2, G1 and G2) during the three experiments (TYR, ION
and FAST).
**Figure 4.** nMDS plot of bacterial community composition over the course of the three experiments
based on Bray-Curtis dissimilarities of 16S rDNA sequences. Samples clustering at different level of
similarity are circled together. All circles represent clusters which are significantly different from each
other ($p < 0.05$) based on a PERMANOVA test.
**Figure 5.** nMDS plot of micro-eukaryotes community composition over the course of the three
experiments based on Bray-Curtis dissimilarities of 18S rDNA sequences. Samples clustering at different
level of similarity are circled together. All circles represent clusters which are significantly different ($p <$
$0.05$) from each other based on a PERMANOVA test.







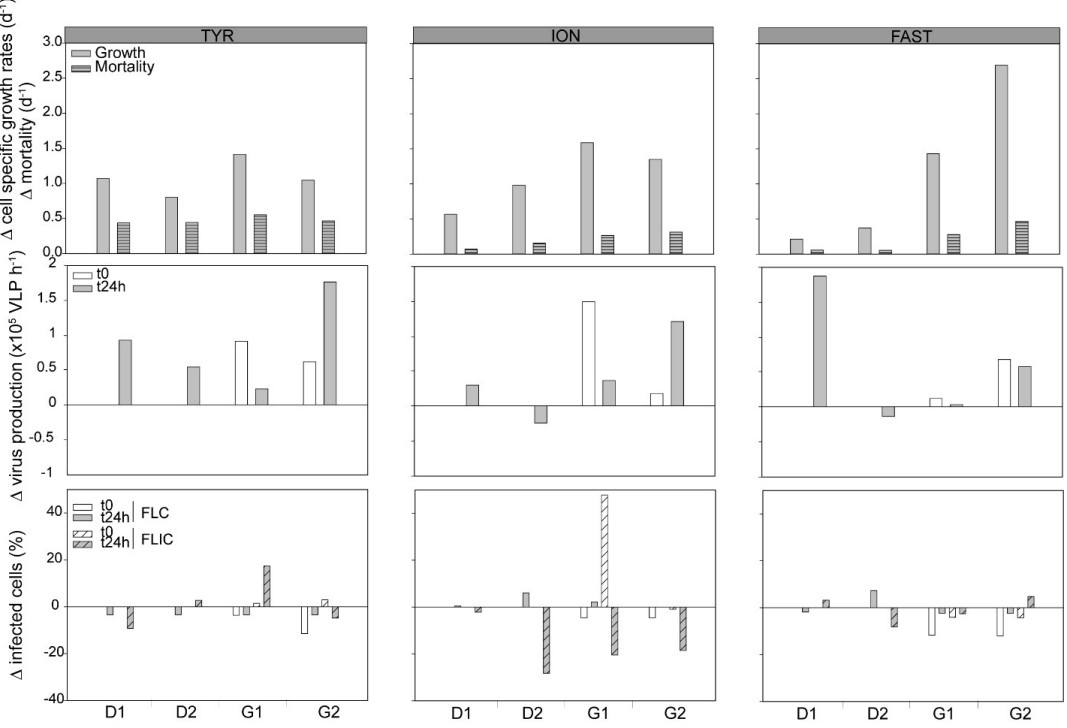


**Figure 1.**





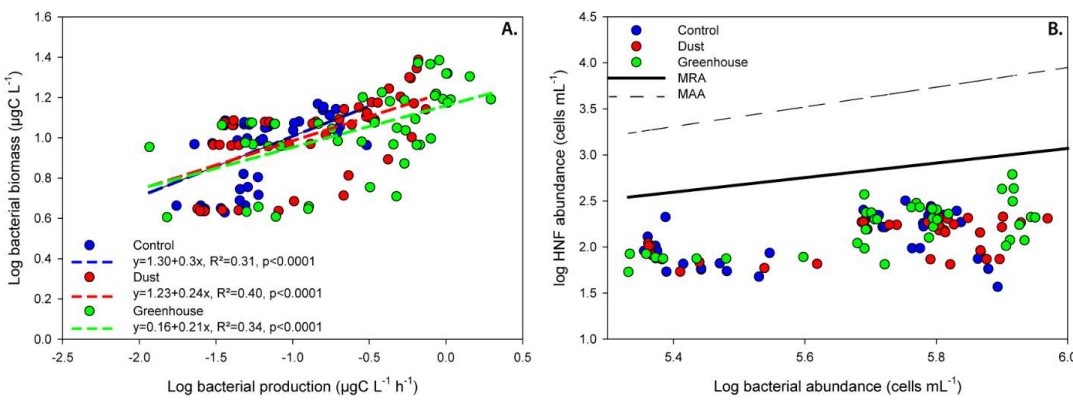

**Figure 2.**






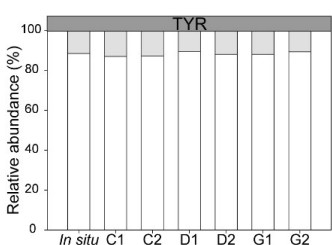 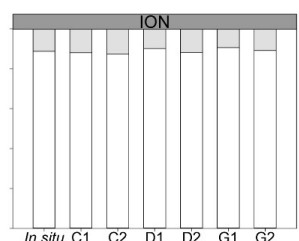 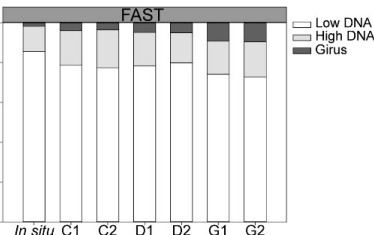

**Figure 3.**



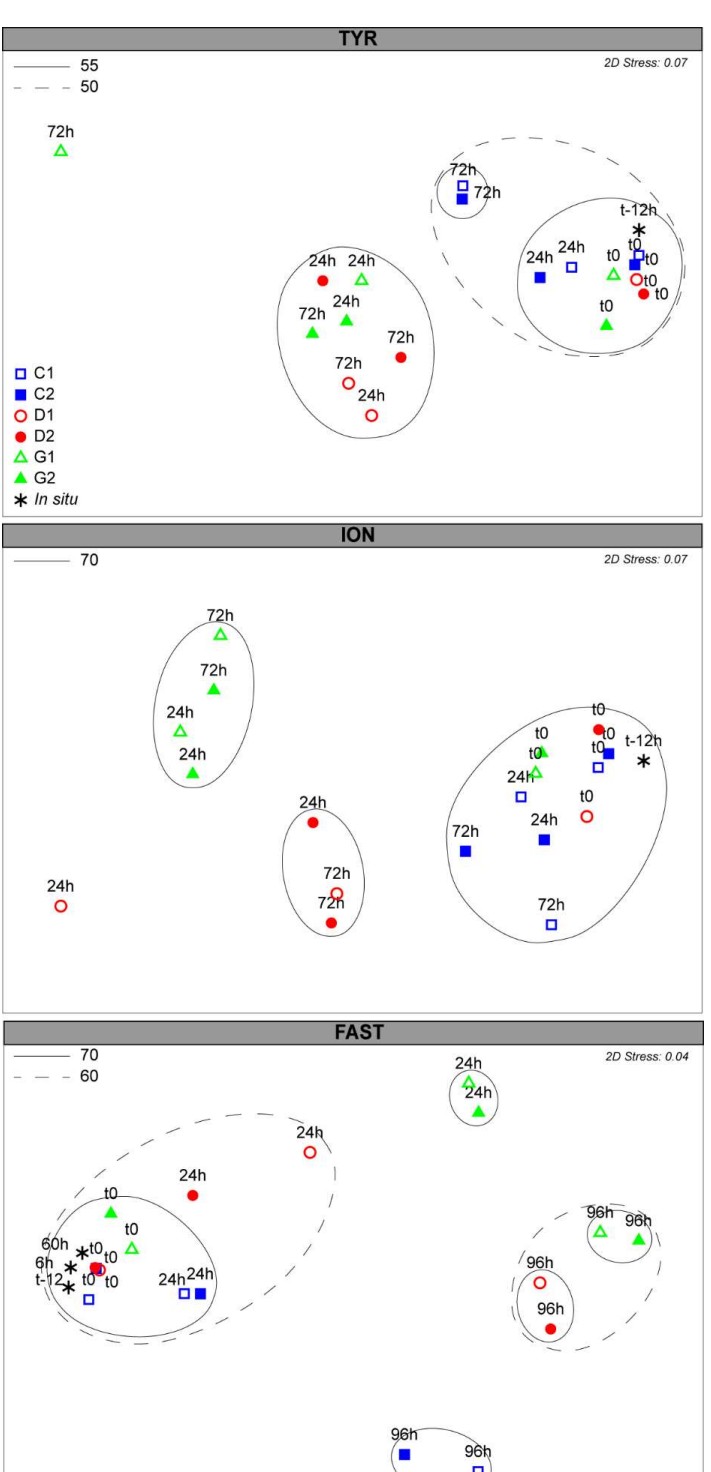

**Figure 4.**





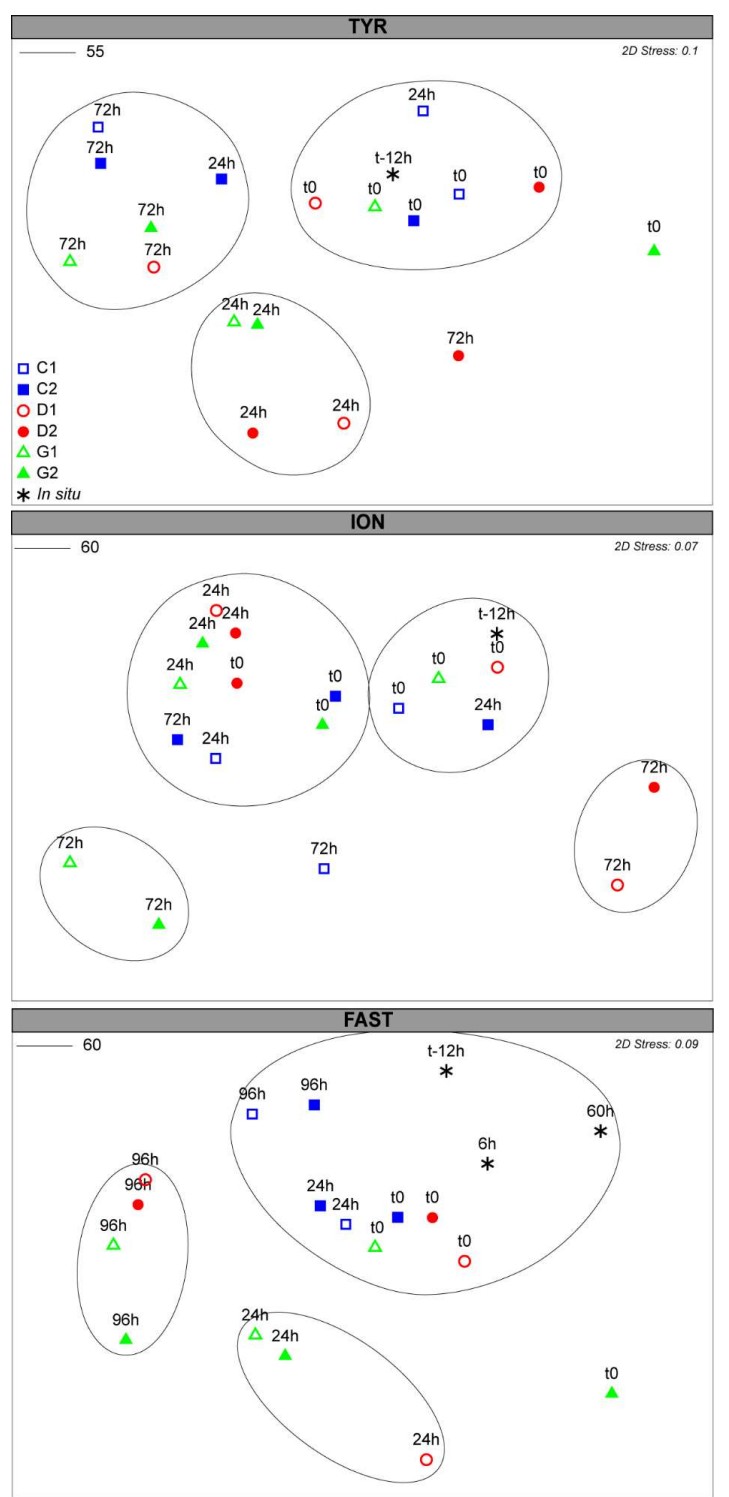

**Figure 5.**