# Peer review of "Impact of dust addition on the microbial food web under present and future conditions of pH and temperature"

_Biogeosciences, 2021_

## Referee Comment (RC1)

**Review of paper by Dinasquet et al. entitled "Impact of dust addition on the microbial food web under present and future conditions of pH and temperature" (MS No.: bg-2021-143).**

Dinasquet et al. studied if and how microbial populations may be affected by dust deposition under present and future (warming and acidification) environmental conditions in 3 basins at the Mediterranean Sea during early summer. This work is part of a much bigger project (PEACETIME).

Results suggest that dust amendments changed the microbial ecosystem from being bottom-up limited to a top-down controlled. These changes are likely attributed to induced viral lysogeny rather than grazing. The authors also suggest that the degree of response of the microbial populations will depend on the initial biogeochemical conditions of the receiving environment. Please see below my comments and suggestions:

1. The focus of the manuscript should go a deep revision to highlight its novelty (that is, viral production and lifestyle aspects following dust amendments) rather than basically repeat the Gazeau et al., work and add on that some other measurements. Currently, this paper is hard to follow without looking at the data provided in Gazeau et al. While I did not read the Gazeau et al., paper (as it is currently under review as I understand it…), from reading its title and the rational provided in lines 311-316 that explains the differences between the studies, I do not really understand the added value of this paper. Thus, the authors should focus more on the viral production and lifestyle aspects which are the most novel, and ignore the rest altogether (which is presented elsewhere). If so, the whole manuscript should be revised accordingly.

2. Seems that much if the results needed to understand what's going on following dust/temperature/pH alterations are, in fact, presented elsewhere (i.e., Gazeau et al.). For example, the temporal dynamics found in microbial variables in the different treatments are not presented at all, but only the change from the control (as delta) in t24 h in Fig.1 (although measurements at t1, t6, t12, t48, t72h were made). And yet, the authors also discuss other time-points, without showing the data at all (many places throughout). This makes it very difficult to assess what

happened in the different minocosms. Contrary, the relative abundance of viral populations is presented and discussed based on the initial vs. t12 h… Please be consistent and present the whole dataset. The way it is presented now is very misleading. Moreover, from reading the discussion I understand that the dust-borne nutrients were measured (possibly also trace-metals, e.g., lines 307-308), however this data is not presented and thus it is hard to see if the changes were triggered by the added 'goods' or by the temperature+pH alterations in treatment G.

Thus, the authors need to show, even in the supporting information, the temporal changes in *Synechococcus*, heterotrophic bacteria, VLP, HNF, BP, nutrients… all the collected data in all time-points. This could be either added as an excel file or as graphs. Otherwise, it is very difficult to comprehend what happened following dust and/or temperature+pH manipulations.

3. The abstract should be revised to better explain what was done, and what were the main results/outcomes. Currently it is very ambiguous and the 'take home message' is unclear. For example, it is unclear which additions (dust or soils? trace-metals/nutrients as imply in Line 35…? etc.) and manipulations (by how much temperature increased? ditto pH) were made. Moreover, the results are vaguely presented (e.g., "mixotrophs were altered", "…Different responses to dust were observed rapidly after addition..." – it's basically says nothing without putting some numbers or more 'direct' explanations… the results suggest that the responses depend on the initial microbial assemblage and metabolic state of the tested water" – how? etc. Were there any differences in responses between basins? All of this should go into a revised abstract.

4. During atmosphere transport the dust particles are typically acidified, which increases micronutrients availability upon deposition in seawater (e.g., Krom et al., 2016). Contrary, in this study, a 'dust analog' was used for the additions rather than dust that passed these atmospheric processes (as in previous studies, e.g., DUNE, Guieu et al., 2010). Therefore, comparing treatment G with treatment D may not be straightforward. Were the same micro- and macronutrients levels were found

between D and G (as the same amount of dust was added in both)? The authors need to discuss this and show the data.

5. In reality, the changes in temperature and pH are gradual and slow (decades), namely they do not occur at once as tested here (minutes to a few hours). Thus, the experimental setup used do not 'allow' microbes to acclimate to these changes, contrary to the 'real world'. I'm wondering how much the results represent the future oceans and the Med Sea. Please discuss this caveat in climate-change studies – this is especially important for bacteria which have faster growth rates than large animals etc.

6. Growth rates and mortality – the authors used an approach with many caveats, uncertainties and uses many assumptions. For example, why do you assume the cells were in exponential phase of growth (Line 115)? Given that marine microbes grow relatively slowly (see review by Kirchman 2016), how can you assume the cells were in exponential growth after only a few hours/day? Moreover, BP is only part of the cellular carbon needs/demand for heterotrophic bacterial cell. I suggest you calculate the bacterial carbon demand, BCD (BP+BP) assuming respiration is ~10% of the total carbon requirements (or alternatively of someone measured respiration that would be ideal), and thus the bacterial growth efficiency (BP/BCD). This may give you a more accurate estimate for heterotrophic growth than just relying on BP.

7. Moreover, the net growth rates were calculated based on three successive sampling points (lines 114-116), but the sampling times were not linear meaning that some points were close to one another (0-1h, 1-6h…) whereas some are daily (24-48 h, 48-72 h…). Were the same time-points used in all treatments for the growth rates and mortality calculations? Are these rates comparable to other reports from LNLC regions?

8.  Some of the methods used should be better described. For example, no information is given on how pico-phytoplankton and heterotrophic bacteria were fixed, processed, which flow cytometer was used, which stain was used for the prokaryote's enumeration (this in contrast to the viruses…). Similarly, how did you measure mitomycin C concentration?

9.  The authors concluded that the initial biogeochemical conditions of the receiving environment (based on oligotrophy? Microbial populations?) are important in understanding the responses of the microbial populations to dust deposition (nowadays and in the future). However, I am not convinced it can be deduced based on only 3 stations (rather than across a nutrient or chlorophyll-a gradient etc.).

10. The conclusion section is a repetition of the discussion and/or refer to other studies from PEACETIME and does not really add much.

Additional comments

| Line 24 | Dust deposition may also have anthropogenic components ('European dust', e.g., Tsagaraki et al., 2017 FMS). |
|---|---|
| Lines 24-25 | There are numerous studies dealing with the influence of dust deposition on microbial processes and community composition, including from the Mediterranean Sea (many of them by the co-authors). |
| Lines 27, 491, 501 | Wet dust deposition sounds like rain mixed with dust. Is that what the authors meant? What is the difference? Is this a technical issue result from the soil's aging (as in Guieu et al., 2010)? Please explain this in mode details in the M&M. |
| Line 35 | It reads like trace metals and nutrients were also manipulated… please revise. |
| Lines 37-38 | How were the mixotrophic community altered? This is a vague description of the results. |
| Lines 33-35 | An ambiguous sentence. |
| Lines 38-40 | "Overall, these results suggest that the effect of dust deposition on the microbial loop is dependent on the initial microbial assemblage and metabolic state of the tested water" – How? Unclear. |
| Lines 48-51 | A very long sentence. Moreover, BP is relatively high (to primary production) in oligotrophic environments such as the Med Sea during summertime. I suggest rephrasing this part. |

| Line 52 | What does it mean "…to this microbial ecosystem…"? Is there other microbial ecosystem in the oceans? |
|---|---|
| Lines 53-55 | The word 'degree' appears twice in the same sentence. |
| Line 56 | If I remember correctly, Ridame's paper showed that dust does not always stimulate N2 fixation (depending on the basin, incubation time, amount added, etc.). |
| Lines 66-68 | The fact that dust events will become more prominent in the future does not necessarily mean that microbial food web might become more dependent on atmospheric deposition of nutrients. You need to better connect with the previous sentence saying that LNLC regions are expanding... enhanced stratification… Currently it reads weird and the flow is not sound. |
| Line 77 | microbial growth and controls (remove the comma). |
| Line 85 | Remove the question mark after "Pourquoi Pas". |
| Line 95 | How much dust was added eventually (in mg/L)? |
| Line 111 | Define HB. |
| Lines 114-118 | Please back up this approach by citing other studies who used it. To me, this approach has many caveats and uses many assumptions that must be discussed. |
| Lines 139-140 | How were the samples preserved before they were run? Did you have an onboard flow cytometer (and thus preservation may not be required)? |
| Line 143 | How mitomycin C was measured? |
| Line 150 | Please give the BS number you used based on the paper cited. |
| Line 192 | Figure 1 does not show the t72h time-point. |
| Lines 192-193 | Were these changes significant? |
| Lines 195-196 | "Heterotrophic bacterial mortality was also higher than in C…" – Which treatment/s? Unclear. |
| Lines 208-209 | You cannot establish a gradient based on 3 points. |
| Line 217 and/or discussion | Please explain what Girus is, and define its size (how was it done FSC?) |
| Lines 222-225 | This is basically true for all variables tested, not only HB viruses' production and life strategy. |
| Lines 238-241 | Which time-point/s? t24? |
| Line 293 | Please explain how May to June are considered 'late spring' (oceanographically-wise) in the Mediterranean area. |
| Lines 299-300 | Bosc et al (line 299) show satellite data and do not present any nutrients data. Ditto D'Ortenzio – it does not present nutrients data but mainly discuss the thermal stability of the water upper column in the Med Sea. Thus, both citations are inappropriate. |
| Line 301 | Define PP. |

| Lines 307-308 | Please show this data. Also – were dust leaching experiments done? If so, how did it differ relative to the values measured in the minicosms? |
|---|---|
| Lines 308-310 | Following what? D or G amendments? Also, which changes? What are you referring to? |
| Lines 396-397 | How do you know? Did you run HPLC analyses and looked for *E. huxleyi* pigment markers? |
| Discussion in subsection 4.2 | Suggested paper to consider – Sharoni et al., (2015). Infection of phytoplankton by aerosolized marine viruses. PNAS. doi/10.1073/pnas.1423667112 |
| Line 443 | The Rahav et al paper is not about dust-borne metals toxicity (unlike Paytan et al., 2009), but on airborne viruses delivered with dust and affect cyanobacterial populations. In fact, this paper should also be considered in subsection 4.2 and/or in lines 461-463. |

---

## Author Response (AR1)

Dear author,

The manuscript presents data complementary to that presented in Gazeau. Unfortunately the presentation of the data seems somewhat arbitrary: It is unclear when the different parameters were estimated and why, in particular when BP and specific growth rates and "net growth rates" were estimated.

*We have removed net growth rates to avoid confusion. The Method section line 120-122 explains when the samples were taken. In the results section we have clarified why we focused on t24h for some of the results, line 206: "Already 24h following dust addition, significant increases in heterotrophic bacterial biomass specific growth rates (BBGR, p ≤ 0.016 at t24h) were observed in all experiments with dust under D and G as seen in Fig. 1 (showing data normalized to C) and Fig. S4. The highest growth rates were observed already 24 h after dust seeding (up to 2.9 d⁻¹ in G2 at FAST, Table S1, Fig.S4)."*

*Therefore, since significant increase compared to controls were observed for BBGR at t24h (now shown in Fig.S4), we estimate mortality rates and viral processes at this time point as presented in Fig 1. We have added table S1 with all the non-normalized data at t24h. We have also added additional data in the supplementary table S1 and figures S2, S3 in response to the reviewers' comments.*

Viral losses were estimated at t0, t24 and T48, but only values at t0 and t24 are shown in Fig. 1 why?

*Line 238: "The sampling strategy for production and life strategies of HB viruses allowed to discriminate independently the effect of i) greenhouse conditions (sampling at T0 before dust addition), ii) dust addition (sampling at t24h) and the combined effects of dust addition and greenhouse."*

*The viral response to dust was immediate already after t24h, to improve clarity we removed t48h.*

Growth rates in Table 2 and Fig 1 (first row) gives values of BP or bacterial accumulation rates? Please define what µapp is in the methods. Figure 1: How are cell specific growth rates determined, are those based on 3H-Leu incorporation or on bacterial accumulation rates?

*Table 2 has been removed to avoid confusion. We have renamed cell specific growth rates to bacterial biomass specific growth rates. We define the rates in the material and methods l.129.*

*"Bacterial biomass specific growth rates (BBGR) were estimated following Kirchman (2002), BP/Bacterial Biomass, assuming a carbon to cell ration of 20 fg C cell⁻¹ (Lee and Fuhrman, 1987)."*

How are VLP (virus production) and bacterial mortality rates estimated? (none of these are mentioned in the materials and methods).

Estimation of virus production (VPL not VLP) is defined in the Material and Method section (Line 141 – 143 in the original manuscript). Details about the method can be found in Weinbauer et al. 2002 and Winter et al. 2004 as referred to in the original manuscript.

In short, please describe all parameters (how they are estimated) and corresponding acronyms used in the materials and methods, and keep those in the result presentation.

Thank you we have made these changes.

It is also difficult to follow when each parameter was measured and why the full dataset is not presented here: it seems that parameters are presented selectively and over different time periods (i.e. Table 2 and Fig. 1), there is, however, no justification for this.

See previous response. We have tried to clarify in the results section and by adding supplementary tables and figures.

Please provide a table with all measurements and their uncertinties carried out here in the manuscript or supplements. Provide rates at comparable time intervals as well as temporal evolution of relevant abundances (i.e. HB, Synechococcus, picoeukaryotes and heterotrohic nanoflagellates (HNF)).

We have added a table in supplementary Table S1.

Further, the discussion seems based more on the evolution of abundances in the treatments (lines 423-459) presented in Gazeau (same issue) without the robust analysis that the parameters presented here deserve; with the available data one would expect robust interpretation of bacterial temporal abundance changes based on growth rates, mortality rates and the contribution of viruses and HNF grazing to mortality (see also reviewers'1 comments).

We do not understand the editor's point. The paper is presenting data that were not in Gazeau et al. In particular the viral processes and the microbial community composition, which are analyzed in depth. The bacterial growth and mortality as well as trophic interactions helps linking this study to the other papers by Gazeau et al.

The trophic interactions are described with the amount of data available in the discussion paragraph 4.1 and viruses processes in 4.2. We have added details to paragraph 4.1 to clarify our discussion.

Please also see response to reviewer 1 and changes made accordingly.

Additional comments:
Line 104: samples for what? BP?

*"Samples for all parameters (except described below) were taken at t-12h (while filling the tanks), t0 (just before dust addition), t1h, t6h, t12h, t24h, t48h, t72h and t96h (after dust addition, and t96h only for FAST)."*

line 111: add "Heterotrophic procaryotes (HB), synechococcus...."

added

Lines 113-116: I imagine here you refer to 2 different parameters? Specific growth rates based on 3H-Leu and accumulation rates ("net growth rates"?) based on evolution of abundances in each treatment at the beginning of the experiments? Does the net accumulation rate correspond to μapp in table 2? Why are the time intervals in Table 2 different?

We removed apparent growth rates to avoid confusion.

line 143:"Briefly, increase in virus abundance in the control tubes represents lytic viral production (VPL), and an increase in mitomycin C treatments represents total (VPT)". Do you mean "Briefly, increase in virus abundance in the control tubes represents lytic viral production (VPL), and in mitomycin C treatments represents total (VPT)". That is for the mitomycin C treatment you also record changes in viral abundances?

Yes the goal of mitomycin C treatment is to induce lysogeny so that all lytic and lysogenic phages are estimated in this treatment.

Line 155 "Briefly, increase in virus abundance in the control tubes represents lytic viral production (VPL), and an increase in treatments with mitomycin C represents total viral production (VPT), *i.e.,* lytic plus lysogenic, viral production."

Fig.1 Legend: replace "raw" with "row".

Corrected

Fig.1: the data in row 1 (specific growth rates) correspond to what sampling time?

Fig1 legend: "The first row represents the bacterial cell specific growth rates and relative mortality rates at t24h after dust addition."

It was also added in the figure legend.

Lines 196-199: how do the regressions in Figure 2 inform on bottom up controls? What is the actual point here? If resources were not limiting no response in dust amended tanks would be expected isn't it?

As mentioned in the discussion the bacterioplankton were nutrient limited in situ. However, in the experimental set-up the bacteria were weakly bottom up controlled in particular in the dust addition treatment, with nutrient input from dust.

Line 201-203: how is top down controls on HNF abundance relevant in the context of this manuscript?

Please see description of these results in the discussion section. This paper is about the microbial loop at large and therefore the HNF dynamics are of interest. It also helps understand the mortality pressure on bacteria. We have added details to this section.

Lines 208-209, Table 1:"The abundance and production of virus-like particles (VLP) increased following an east to west gradient (Table 1).". It would make sense to organise stations in the table also from east to west." Also please indicate in the text: "The initial abundances and production of virus-like...".

This was corrected following reviewers comments to "The initial abundance and production of virus-like particles (VLP) was higher in the western stations (Table 1)."

Lines 209-212: FLC and FLIC show actually opposite trends which are unrelated to VLP and Virus production rates.

FLIC and FLC are the frequency of lytically infected cells and the frequency of lysogenic cells respectively. They are dependent on both the viral production and the bacterial abundance (HB) as described in the material and method:

FLIC = 100 x VPL / BS x $HB_i$ (1)
FLC = 100 x VPLG / BS x $HB_i$ (2)
where $HB_i$ is the initial HB abundance in the viral production experiment and BS is a theoretical burst size of 20 viruses per infected cell (averaged BS in marine oligotrophic waters, Parada et al., 2006).

Line 213-222: In terms of understanding processes Figure S3 would be more relevant. I would suggest providing Fig. S3 with the main text, and explain changes in proportion based on the evolution of abundances.

Fig S3 was moved to the main manuscript as Fig. 4 as suggested.

Lines 318-314: The manuscript discusses top-down and bottom-up controls using indexes without referring to the rates presented i.e. differences between bacterial production and accumulation rates, or simply the evolution of bacterial abundances in dust amended vs. control are the best indicators of top-down/bottom-up control.

The evolution of bacterial abundance alone only shows us that nutrients limitation at the stations could be alleviated by dust addition.

However here we show that there's potential additional top down controls on the community and that may explain some elevated mortality rates. Which we further try to assign to viral or grazing pressure. However, without additional experiments such as grazing experiments, we cannot assess directly which are the mortality agents.

Lines 496-498:"Overall, the impact of such simulated pulsed nutrient deposition will depend on the initial biogeochemical conditions of the ecosystem, with likely possible large impact on microbial trophic interactions and community structure." It is not clear how the data presented here supports this conclusion.

The data presented here are showing variations in prokaryotes and eukaryotes community structures and trophic interactions through changes in viral processes in particular. We have modified to: *"Overall, the impact of such simulated pulsed nutrient deposition will depend on the initial biogeochemical conditions of the ecosystem, with likely possible large impact on microbial trophic interactions, in particular viral processes, and community structure."*

Sincerely,
Christine Klaas

**Response to referee RC1: Dinasquet et al. Impact of dust addition on the microbial food web under present and future conditions of pH and temperature - bg2021-143.**

**https://bg.copernicus.org/preprints/bg-2021-143/bg-2021-143.pdf**

Dinasquet et al. studied if and how microbial populations may be affected by dust deposition under present and future (warming and acidification) environmental conditions in 3 basins at the Mediterranean Sea during early summer. This work is part of a much bigger project (PEACETIME). Results suggest that dust amendments changed the microbial ecosystem from being bottom-up limited to a top-down controlled. These changes are likely attributed to induced viral lysogeny rather than grazing. The authors also suggest that the degree of response of the microbial populations will depend on the initial biogeochemical conditions of the receiving environment. Please see below my comments and suggestions:

1. The focus of the manuscript should go a deep revision to highlight its novelty (that is, viral production and lifestyle aspects following dust amendments) rather than basically repeat the Gazeau et al., work and add on that some other measurements. Currently, this paper is hard to follow without looking at the data provided in Gazeau et al. While I did not read the Gazeau et al., paper (as it is currently under review as I understand it…), from reading its title and the rational provided in lines 311-316 that explains the differences between the studies, I do not really understand the added value of this paper. Thus, the authors should focus more on the viral production and lifestyle aspects which are the most novel, and ignore the rest altogether (which is presented elsewhere). If so, the whole manuscript should be revised accordingly.

*We understand that the reviewer may be confused on the scope of the present study, without reading the two companion papers. These two papers were available as preprint before our submission in the same special issue. The first paper Gazeau et al. (2020) in the manuscript is now published and now referred to as Gazeau et al. (2021a). The second paper Gazeau et al. (2021b) is currently in press.*

*Briefly Gazeau et al. (2021a), is the introduction paper to this experiment, it describes the experimental overview: the complex set-up of the experiment, and the state-of-the-art minicosm tanks used. In this paper the main variables: e.g. nutrients, temperature, pH as well as the biological stocks (e.g. pigments and cytometry data) were presented.*

*In Gazeau et al 2021a the other manuscripts associated to this experiment are also presented: "Other manuscripts, related to these experiments in this special issue, focus on plankton metabolism (primary production, heterotrophic prokaryote production) and carbon export (Gazeau et al., 2021b), microbial food web (Dinasquet et al., 2021), nitrogen fixation (Ridame et al., 2021) and on the release of insoluble elements (Fe, Al, REE, Th, Pa) from dust (Roy-Barman et al., 2021)."*

*In Gazeau et al. (2021b), the authors present the impact of dust on plankton metabolism and carbon export during this experiment.*

*The present paper, Dinasquet et al., is a follow up of these two papers, zooming onto the microbial loop, which has not been presented elsewhere. In this manuscript, we describe mortality rates and potential bottom-up and top-down controls (including viral processes), as well as community composition.*

2. Seems that much if the results needed to understand what's going on following dust/temperature/pH alterations are, in fact, presented elsewhere (i.e., Gazeau et al.). For example, the temporal dynamics found in microbial variables in the different treatments are not presented at all, but only the change from the control (as delta) in t24 h in Fig.1 (although measurements at t1, t6, t12, t48, t72h were made). And yet, the authors also discuss other time-points, without showing the data at all (many places throughout). This makes it very difficult to assess what happened in the different minocosms. Contrary, the relative abundance of viral populations is presented and discussed based on the initial vs. t12 h… Please be consistent and present the whole dataset. The way it is presented now is very misleading. Moreover, from reading the discussion I understand that the dust-borne nutrients were measured (possibly also trace-metals, e.g., lines 307-308), however this data is not presented and thus it is hard to see if the changes were triggered by the added 'goods' or by the temperature+pH alterations in treatment G. Thus, the authors need to show, even in the supporting information, the temporal changes in Synechococcus, heterotrophic bacteria, VLP, HNF, BP, nutrients… all the collected data in all time-points. This could be either added as an excel file or as graphs. Otherwise, it is very difficult to comprehend what happened following dust and/or temperature+pH manipulations.

*As discussed above, this is a companion paper presenting the microbial loop. We are interpreting some of the data already presented in Gazeau et al. (2021a,b), in the same special issue, for the context of this paper, and are presenting only the data that are novel and essential to the paper. However, to avoid any confusions, we have added figures from Gazeau et al. (2021a) (Figures 6 nutrients and 8 microbial abundances) and Gazeau et al. (2021b) (Figure 7, bacterial production), showing the results of importance to the present study as supplementary Fig. S2 and S3. We have also added the absolute values at T24h as Table S1 to complement Fig. 1.*

*We have added to the results the following statements: "Nutrients inputs were observed with dust addition (Fig.S2) and in response the autotrophic and heterotrophic microbes abundance increased, as well as bacterial production in both D and G treatments (Fig. S3), as described in more details in Gazeau et al. (2021a,b)."*

*"Already after 24h, in both D and G, heterotrophic bacterial mortality rates were higher than in C, especially at TYR in D (up 0.5 $d^{-1}$) and in G at ION (up to 0.6 $d^{-1}$) and FAST (up to 0.7 $d^{-1}$) (Fig. 1, Table S1)."*

3. The abstract should be revised to better explain what was done, and what were the main results/outcomes. Currently it is very ambiguous and the 'take home message' is unclear. For example, it is unclear which additions (dust or soils? tracemetals/nutrients as imply in

Line 35…? etc.) and manipulations (by how much temperature increased? ditto pH) were made. Moreover, the results are vaguely presented (e.g., "mixotrophs were altered", "…Different responses to dust were observed rapidly after addition..." – it's basically says nothing without putting some numbers or more 'direct' explanations… the results suggest that the responses depend on the initial microbial assemblage and metabolic state of the tested water" – how? etc. Were there any differences in responses between basins? All of this should go into a revised abstract.

*We have made some changes to the abstract following the reviewer's suggestions. Also, since the experiment was so complex, we think it would make the abstract too long to add too many details to the results (variable, Basin, treatments…)*

4. During atmosphere transport the dust particles are typically acidified, which increases micronutrients availability upon deposition in seawater (e.g., Krom et al., 2016). Contrary, in this study, a 'dust analog' was used for the additions rather than dust that passed these atmospheric processes (as in previous studies, e.g., DUNE, Guieu et al., 2010). Therefore, comparing treatment G with treatment D may not be straightforward.

*Description of the dust analog composition sampling and chemical processing is provided in detail in Gazeau et al. (2021a):*

*"The same dust analog flux was applied as in the DUNE 2009 experiments described in Desboeufs et al. (2014). The dust was derived from the <20 μm fraction of soil collected in Southern Tunisia (a major source for material transported and deposited in the Northwestern Mediterranean) consisting of quartz (40%), calcite (30%) and clay (25%) with most particles (99%) smaller than 0.1 μm (Desboeufs et al., 2014). The collected material underwent an artificial chemical aging process by addition of nitric and sulfuric acid ($HNO_3$ and $H_2SO_4$, respectively) to mimic cloud processes during atmospheric transport of aerosol with anthropogenic acid gases (Guieu et al., 2010a, and references therein). To mimic a realistic wet flux event for the Mediterranean of 10 g m$^{-2}$, 3.6 g of this analog dust were quickly diluted in 2 L ultrahigh-purity water (UHP water; 18.2 MΩ cm$^{-1}$ resistivity), and sprayed at the surface of the tanks using an all-plastic garden sprayer (duration = 30 min). The total N and P mass in the dust were $1.36 \pm 0.09\%$ and $0.055 \pm 0.003\%$, respectively (see Desboeufs et al., 2014, for a full description of dust chemical composition)."*

*"(…) based on previous studies reporting the mixing between dust and polluted air masses during the atmospheric transport of dust particles (e.g. Falkovich et al., 2001; Putaud et al., 2004), we used an evapo-condensed dust analog that mimics the processes taking place in the atmosphere prior to deposition, essentially the adsorption of inorganic and organic soluble species (e.g. sulfate and nitrate; see Guieu et al., 2010a, for further details)."*

*We have added parts of this description in the manuscript M&M as:* *"The same dust analog was used as during the DUNE 2009 experiments described in Desboeufs et al. (2014) and the same dust wet flux of 10 g m$^{-2}$ was simulated (as described in Gazeau et al., 2021a). Briefly, the dust was derived from the <20 μm fraction of soil collected in Southern Tunisia (a major source for material transported and deposited in the Northwestern Mediterranean) with most particles*

(99%) smaller than 0.1 μm (Desboeufs et al., 2014). The collected material underwent an artificial chemical aging process by addition of nitric and sulfuric acid (HNO$_3$ and H$_2$SO$_4$, respectively) to mimic cloud processes during atmospheric transport of aerosol with anthropogenic acid gases (Guieu et al., 2010, and references therein). To mimic a realistic wet flux event for the Mediterranean of 10 g m$^{-2}$, 3.6 g of this analog dust were quickly diluted in 2 L ultrahigh-purity, and sprayed at the surface of the dust amended treatments (D1, D2 and G1, G2; Gazeau et al., 2021a). Such deposition event represents a high but realistic scenario, as several studies reported even higher short wet deposition events in this area of the Mediterranean Sea (Ternon et al., 2010; Bonnet and Guieu, 2006; Loÿe-Pilot and Martin, 1996), suggesting that wet deposition is the main pathway of dust input in the Western Mediterranean Sea."

Were the same micro- and macronutrients levels were found between D and G (as the same amount of dust was added in both)? The authors need to discuss this and show the data.

*The nutrient data are presented in Gazeau et al. (2021a), but we have added Fig S2 (Figure 6 from Gazeau et al., 2021a) presenting the amount of nutrient added following dust seeding.*

4. In reality, the changes in temperature and pH are gradual and slow (decades), namely they do not occur at once as tested here (minutes to a few hours). Thus, the experimental setup used do not 'allow' microbes to acclimate to these changes, contrary to the 'real world'. I'm wondering how much the results represent the future oceans and the Med Sea. Please discuss this caveat in climate-change studies – this is especially important for bacteria which have faster growth rates than large animals etc.

*Indeed, while seawater temperature was increased during the night prior to the start of the experiment (dust seeding after T0), acidification was performed by addition of CO2-saturated filtered seawater at once to decrease the pH by 0.3 pH units. We agree with the referee that organisms were not acclimated to these levels of temperature and pH prior to dust addition. As stated in the introduction of Gazeau et al. (2021a), the objectives of this experiments were not to study acidification and temperature effects per se, but rather to study whether plankton will react differently to dust deposition in a warmer and acidified environment. Based on several studies that we conducted in the past:*

*Maugendre et al. (2017) - L. Maugendre, C. Guieu, J.-P. Gattuso, F. Gazeau, Ocean acidification in the Mediterranean Sea: Pelagic mesocosm experiments. A synthesis, Estuarine, Coastal and Shelf Science, Volume 186, Part A, Pages 1-10, https://doi.org/10.1016/j.ecss.2017.01.006. (https://www.sciencedirect.com/science/article/pii/S0272771417300124)*

*Maugendre et al. (2015) - L. Maugendre, J.-P. Gattuso, J. Louis, A. de Kluijver, S. Marro, K. Soetaert, F. Gazeau, Effect of ocean warming and acidification on a plankton community in the NW Mediterranean Sea, ICES Journal of Marine Science, Volume 72, Issue 6, July/August 2015, Pages 1744–1755, https://doi.org/10.1093/icesjms/fsu161)*

*, we are very confident that under nutrient depletion, as this was the case for all three experiments during our study, these two environmental changes (warming by 3°C and acidification by -0.3 pH units) did not exert a significant control on the communities prior to dust seeding and associated relieving of nutrient limitation.*

*This is the reason why 1) we did not consider a treatment to study acidification and warming without dust seeding and 2) why we did not expose the communities to these changes for a longer period prior to dust seeding. We strongly believe that this would have created an even larger bias to our experiments related to confinement issues.*

5. Growth rates and mortality – the authors used an approach with many caveats, uncertainties and uses many assumptions. For example, why do you assume the cells were in exponential phase of growth (Line 115)? Given that marine microbes grow relatively slowly (see review by Kirchman 2016), how can you assume the cells were in exponential growth after only a few hours/day? Moreover, BP is only part of the cellular carbon needs/demand for heterotrophic bacterial cell. I suggest you calculate the bacterial carbon demand, BCD (BP+BP) assuming respiration is ~10% of the total carbon requirements (or alternatively of someone measured respiration that would be ideal), and thus the bacterial growth efficiency (BP/BCD). This may give you a more accurate estimate for heterotrophic growth than just relying on BP.

*The variations in bacterial growth efficiency are presented in Gazeau et al. (2021b) which discuss metabolic balance in the minicosms. This is not the object of the present study. Rather, we present BP, not BCD, as only production of Hprok cells our Hprok organic carbon can be compared to lysis or grazing fluxes to estimate the fate of heterotrophic prokaryotes through the microbial food web.*

6. Moreover, the net growth rates were calculated based on three successive sampling points (lines 114-116), but the sampling times were not linear meaning that some points were close to one another (0-1h, 1-6h…) whereas some are daily (24-48 h, 48-72 h…). Were the same time-points used in all treatments for the growth rates and mortality calculations?

*The exponential growth phase was detected after plotting the data in ln scale, in general the regression included at least 3 time points and the corresponding data are presented table 2. Note that for a given parameter the period considered (and hence its duration) is mostly the same from one minicosm to another and as such the data are comparable.*

*For heterotrophic prokaryotes, we assumed that BP is a better proxy than abundance of HB to search for exponential growth phase, because all cells are not actively growing. BP increase was immediate and exponential (12h after dust addition). Thus, samples T0, T1h T6h T12h were used to plot the regression, whereas in general, growth of Synechococcus and picoeukaryotes cells was delayed and occurred between T12h and T48h (the regression is based on T12h, T24h and T48h data points). Because the net evolution of abundances is always the balance between*

*actual growth and mortality sources, we called all the data presented table 2 "apparent growth rates."*

*However, to avoid confusion we moved table 2 as supplementary table S2 and removed bacterial net growth rates to only focus on bacterial biomass specific growth rates (Table S1 and Fig 1)*

*For comparison with mortality rates, we calculated "instantaneous" biomass specific growth rates at each sampling time point using the following equation:*

*Biomass specific growth rate = Bacterial Production/Bacterial Biomass (Kirchman 2002)*

*Assuming 20 fgC cell$^{-1}$ (to be consistent with Gazeau et al 2021a,b; this was clarified in the M&M).*

*This "instantaneous" biomass specific growth rates can be calculated at each time point and is thus directly comparable to the mortality rates calculated at time 0, 24 and 48h. Note, however, that it has the advantage to give instantaneous values at each time point, but it is biased by the heterogeneity of population growth rates and cell to carbon biomass conversion factor*

*Reference added: Kirchman 2002 'Calculating microbial growth rates from data on production and standing stocks' MEPS 233 :303-306*

- Are these rates comparable to other reports from LNLC regions?

*Only initial samples (T-1, before confinement in minicosm and dust addition) represent the instantaneous biomass specific growth rates in situ and could be compared to reports from HNLC regions. Such values were 0.03, 0.08 and 0.07 d$^{-1}$ at TYR, ION and FAST, respectively, i.e. in the range of values (BP/BB) found in temperate regions of 0.03 – 0.3 d$^{-1}$(Ducklow, 2000; Kirchman, 2016)*

8. Some of the methods used should be better described. For example, no information is given on how pico-phytoplankton and heterotrophic bacteria were fixed, processed, which flow cytometer was used, which stain was used for the prokaryote's enumeration (this in contrast to the viruses…).

*All these methods are described in detail in Gazeau et al 2021b, in the same special issue. A brief description was added to the manuscript:*

'Briefly, samples (4.5 mL) were fixed with glutaraldehyde grade I (1% final concentration) and stored at -80°C until analysis.  Counts were performed on a FACSCanto II flow cytometer (Becton Dickinson©) following Marie et al. (2010) for autotrophic cells. For the enumeration of heterotrophs (bacteria and HNF), cells were stained with SYBR Green I at 0.025% (vol / vol) final concentration (Gasol & DelGiorgio, Christaki et al 2011)."

- Similarly, how did you measure mitomycin C concentration?

*The reviewer probably misunderstood the protocol for lysogeny induction. The commercial stock of mitomycin C (known concentration) was directly diluted in our sample at a final concentration of 1µg mL⁻¹. We clarified this as:*

*"Briefly, increase in virus abundance in the control tubes represents lytic viral production (VPL), and an increase in treatments with mitomycin C represents total viral production (VPT), i.e., lytic plus lysogenic, viral production."*

9. The authors concluded that the initial biogeochemical conditions of the receiving environment (based on oligotrophy? Microbial populations?) are important in understanding the responses of the microbial populations to dust deposition (nowadays and in the future). However, I am not convinced it can be deduced based on only 3 stations (rather than across a nutrient or chlorophyll-a gradient etc.).

*To our knowledge, this is the first attempt to run dust deposition experiments under the same protocol and using the same dust enrichment in three different areas, which had different initial metabolic and microbial community composition. This is discussed in Gazeau et al 2021a,b where the initial contrasted biogeochemical status, microbial abundances and metabolic balance at FAST, ION and TYR are described in detail.*

10. The conclusion section is a repetition of the discussion and/or refer to other studies from PEACETIME and does not really add much.

*We disagree with the reviewer and think it is a good addition to the paper for reader that do not plan to read the full discussion and want to know more by reading the companion studies in this special issue.*

Additional comments

Line 24 Dust deposition may also have anthropogenic components ('European dust', e.g., Tsagaraki et al., 2017 FMS).

*Revised to "deposition of aerosols from both natural (e.g. Saharan dust), anthropogenic or mixed origins."*

Lines 24- 25 There are numerous studies dealing with the influence of dust deposition on microbial processes and community composition, including from the Mediterranean Sea (many of them by the co-authors).

*We agree with the reviewer, but we implied that despite research we still do not yet fully understand the processes.*

Lines 27, 491, 501 Wet dust deposition sounds like rain mixed with dust. Is that what the authors meant? What is the difference? Is this a technical issue result from the soil's aging (as in Guieu et al., 2010)? Please explain this in mode details in the M&M.

*The objective of this study was to study the impact of wet dust deposition, which is the main dust deposition pathway in the Western Mediterranean Sea (Loÿe-Pilot & Martin 1996). We rephrased this in the M&M as previously mentioned in response to comment 4.*

Line 35 It reads like trace metals and nutrients were also manipulated… please revise.

*Revised to "The dust input of nutrients and trace metals"*

Lines 37- 38 How were the mixotrophic community altered? This is a vague description of the results.

*Revised to "The composition of mixotrophic microeukaryotes and phototrophic prokaryotes increased."*

Lines 33- 35 An ambiguous sentence.

*Revised to "Different microbial and viral responses to dust were observed rapidly after addition and were most of the time more pronounced when combined to future environmental conditions."*

Lines 38- 40 "Overall, these results suggest that the effect of dust deposition on the microbial loop is dependent on the initial microbial assemblage and metabolic state of the tested water" – How? Unclear.

*The sentence is grammatically correct, we don't want to expand on "how" in the abstract*

Lines 48- 51 A very long sentence. Moreover, BP is relatively high (to primary production) in oligotrophic environments such as the Med Sea during summertime. I suggest rephrasing this part.

*The sentence was cut in 2 as:*

*'The summer Mediterranean food web is characterized by low primary production (PP) and heterotrophic prokaryotic production (more classically abbreviated as BP for bacterial production) constrained by nutrient availability. Low BP further limits dissolved organic matter (DOM) utilization and export, resulting in DOM accumulation.'*

Line 52 What does it mean "…to this microbial ecosystem…"? Is there other microbial ecosystem in the oceans?

*The sentence was modified as:*

*'Therefore, inputs of bioavailable nutrients through deposition of atmospheric particles are essential to the Mediterranean Sea microbial ecosystem'*

Lines 53- 55 The word 'degree' appears twice in the same sentence.

*The sentence was modified as:*

*'Indeed, these nutrient pulses have been shown to support microbial processes but the extent to which the microbial food web is affected might be dependent on the degree of oligotrophy of the water (Marín-Beltrán et al., 2019; Marañon et al., 2010).'*

Line 56 If I remember correctly, Ridame's paper showed that dust does not always stimulate N2 fixation (depending on the basin, incubation time, amount added, etc.).

*The sentence was modified and the reference to Ridame et al 2021 (preprint in this special issue, showing impact of dust on N2 fixation) has been added:*

'In the Mediterranean Sea, dust deposition may stimulates PP and $N_2$ fixation (Guieu et al., 2014; Ridame et al., 2011, 2021)'

Lines 66- 68 The fact that dust events will become more prominent in the future does not necessarily mean that microbial food web might become more dependent on atmospheric deposition of nutrients. You need to better connect with the previous sentence saying that LNLC regions are expanding... enhanced stratification… Currently it reads weird and the flow is not sound.

*The sentence was modified as:*

'For these reasons, in the future ocean, the microbial food web might become even more dependent on atmospheric deposition of nutrients'.

Line 77 microbial growth and controls (remove the comma).

Revised

Line 85 Remove the question mark after "Pourquoi Pas".

This is the complete name of the research vessel, with the question mark!

Line 95 How much dust was added eventually (in mg/L)?

*This was described in the lines below 98-101 of the submitted manuscript:*

'dust wet flux of 10 g m$^{-2}$ was simulated. Such deposition event represents a high but realistic scenario, as several studies reported even higher short deposition events in this area of the Mediterranean Sea (Ternon et al., 2010; Bonnet and Guieu, 2006; Loÿe-Pilot and Martin, 1996). After mixing the dust analog (3.6 g) in 2 L of ultrahigh-purity water, this solution was sprayed at the surface of the dust amended treatments'

*According to the minicosm surface area of 0.36 m$^2$ and total seawater volume of 280 L, the volumetric concentration of dust was 12.8 mg L$^{-1}$*

This is now rephrased in material and methods sections as described in response to comment 4.

Line 111 Define HB.

*Heterotrophic prokaryotes*

Lines 114- 118 Please back up this approach by citing other studies who used it. To me, this approach has many caveats and uses many assumptions that must be discussed.

*See response to general comment 6*

Lines 139- 140 How were the samples preserved before they were run? Did you have an onboard flow cytometer (and thus preservation may not be required)?

*The virus protocol is described at the beginning of paragraph 2.3. We also added a brief description of the flow cytometry protocol for auto- and heterotrophs (as mentioned previously).*

Line 143 How mitomycin C was measured?

*We meant an increase in the treatment where mitomycin C was added and thus lysis induced. This is described in comment 8.*

Line 150 Please give the BS number you used based on the paper cited.

*It was already written in the sentence: '20 viruses per infected cells'.*

Line 192 Figure 1 does not show the t72h time-point.

*The sentence was modified as:*

Significant increases in heterotrophic bacterial cell specific growth rates ($p \leq 0.016$ after 24 h and 72 h) were observed in all experiments with dust under D and G relative to C, the highest growth rates relative to C were observed already 24 h after dust seeding (up to 2.7 $d^{-1}$ in G2 at FAST, Fig. 1).

Lines 192- 193 Were these changes significant?

*Yes we presented statistical data in the sentence in the submitted version.*

Lines 195- 196 "Heterotrophic bacterial mortality was also higher than in C…" – Which treatment/s? Unclear.

*This was revised as:*

"Already after 24h, in both D and G, heterotrophic bacterial mortality rates were higher than in C, especially at TYR in D (up 0.5 $d^{-1}$) and in G at ION (up to 0.6 $d^{-1}$) and FAST (up to 0.7 $d^{-1}$) (Fig. 1, Table S1)"

Lines 208- 209 You cannot establish a gradient based on 3 points.

*The sentence was modified as:*

'The abundance and production of virus-like particles (VLP) was higher in the western stations (Table 1)'

Line 217 and/or discussion Please explain what Girus is, and define its size (how was it done FSC?)

*This was described in the M&M and discussion and Fig.S1. We have added the definition of Girus (Giant Virus) in the text and in Fig S1. We also provided a more detailed description of Figure S1 to help interpret the flow cytogram.*

**"Figure S1:** Determination of the viral populations by flow cytometry. Three main viral populations were discriminated based on their DNA fluorescence (DNA axis) and Side Scatter

(SSC axis). The population of Low DNA viruses generally comprises viruses of bacteria (phages) with small genome, that of High DNA viruses is made of viruses with larger genome size (generally 200 – 300 kb) such as for some viruses of cyanobacteria or picoeukaryotes) while the Girus (giant virus) population typically comprises viruses of nanoeukaryotes (*e.g.*, microalgae, HNF) with giant genome (generally > 300 kb).”

Lines 222- 225 This is basically true for all variables tested, not only HB viruses' production and life strategy.

*No this is the only instance where incubation in the Greenhouse G conditions are done independent of dust addition, because the water which was subsampled for 18h incubation to study virus processes was sampled before the addition of dust*

Lines 238- 241 Which time-point/s? t24?

*This statement implies at some point during the experiment, the details are described in the following paragraphs.*

Line 293 Please explain how May to June are considered 'late spring' (oceanographically-wise) in the Mediterranean area.

*Please read the sentences following this in the text (and our response to the next comment below).*

Lines 299- 300 Bosc et al (line 299) show satellite data and do not present any nutrients data. Ditto D'Ortenzio – it does not present nutrients data but mainly discuss the thermal stability of the water upper column in the Med Sea. Thus, both citations are inappropriate.

*No but they describe the stratification which is the statement we are referring to (explaining the late spring/early summer conditions) since we have measured the nutrients during the present study (see Gazeau et al. 2021a and Guieu et al 2020).*

*We have cut the sentence in two to avoid confusion:* “Briefly, very low levels of dissolved inorganic nutrients were measured at all three stations, highlighting the oligotrophic status of the waters. This is typical of the stratified conditions generally observed in the Mediterranean Sea in late spring/early summer (*e.g.,* Bosc et al., 2004; D'Ortenzio et al., 2005).”

Line 301 Define PP.

*It is primary production, which was already defined in the introduction.*

Lines 307- 308 Please show this data.

*See general comment 1, all of this is described in the cited companion paper Gazeau et al 2021a*

- Also – were dust leaching experiments done? If so, how did it differ relative to the values measured in the minicosms?

Similar N, P enrichment were observed during abiotic experiments run with the same dust analog: Louis, J., Gazeau, F., & Guieu, C. (2018). Atmospheric nutrients in seawater under current and high pCO2 conditions after Saharan dust deposition: Results from three minicosm experiments. *Progress in Oceanography*, *163*, 40-49.

Lines 308- 310 Following what? D or G amendments? Also, which changes? What are you referring to?

*See general comment 1, this is following dust addition in both D and G minicosms and described in the previous sentence. Also this is described in details in Gazeau et al 2021a,b. and now added in Fig. S2, S3. The sentence has been modified as:*

"Rapid changes were observed on plankton stocks (autotrophs and heterotrophs abundance and chl.a, Gazeau et al., 2021a) and metabolisms (BP and PP, Gazeau et al., 2021b), suggesting that the impact of dust deposition is constrained by the initial composition and metabolic state of the investigated community."

Lines 396- 397 How do you know? Did you run HPLC analyses and looked for E. huxleyi pigment markers?

*This is described in the present study with the 18S rDNA results (also we did run HPLC for pigments analysis, see Gazeau et al 2021a), see table S2, results 3.3.2 and discussion line 450-453 of the submitted manuscript.*

Discussion in subsection 4.2 Suggested paper to consider – Sharoni et al., (2015). Infection of phytoplankton by aerosolized marine viruses. PNAS. doi/10.1073/pnas.1423667112

*We added this reference:*

*'Aerosol deposition was already identified as a factor that stimulates virus production and viral induced mortality of bacteria in the Mediterranean Sea (Pulido-Villena et al., 2014; Tsiola et al., 2017) and direct deposition of airborne viruses and viruses attached to dust particles may also affect microbial food webs (Sharoni et al., 2015; Rahav et al., 2020).'*

Line 443 The Rahav et al paper is not about dust-borne metals toxicity (unlike Paytan et al., 2009), but on airborne viruses delivered with dust and affect cyanobacterial populations.

*This was revised to add the deposition of biological particles:*

'Potential toxicity effects of metals and biological particles released from dust/aerosols on certain micro-organisms have also been reported (Paytan et al., 2009; Rahav et al., 2020)'.

*Reference was also added to the statement "Positive to toxic impacts on cyanobacteria have been reported from atmospheric deposition experiments (e.g., Paytan et al., 2009; Zhou et al., 2021, Rahav et al., 2020)"*

In fact, this paper should also be considered in subsection 4.2 and/or in lines 461-463.

*Both suggested publications were added to the discussion in subsection 4.2*

**Response to referee RC2: Dinasquet et al. Impact of dust addition on the microbial food web under present and future conditions of pH and temperature - bg2021-143.**

**https://bg.copernicus.org/preprints/bg-2021-143/bg-2021-143.pdf**

General comments

The manuscript "Impact of dust addition on the microbial food web under present and future conditions of pH and temperature" by Dinasquet et al. investigated atmospheric wet dust deposition impacted on the microbial food web under warming and acidification environmental conditions in 300 L climate reactors. The authors found that the effect of dust deposition on the microbial loop is dependent on the initial microbial assemblage and metabolic state of the tested water, and that predicted warming, and acidification will intensify these responses, affecting food web processes and biogeochemical cycles. This manuscript addresses interesting scientific issues and was generally well written with meaningful results. Some minor issues are listed below for further improvements for publication.

*We appreciate the reviewer interest and thank them for the suggestions.*

Specific comments

L97, how did the authors maintain the constant pH in the incubations, did you measure it over the time?

*pH was measured throughout the experiment onboard and is presented in Gazeau et al. (2021a). pCO$_2$ enriched air was flushed above the tanks, and only moderate pH increases were observed over time. We added a sentence in the M&M to briefly describe acidification and warming of the water prior to the dust addition as:* "Water was acidified by addition of CO$_2$ saturated 0.2 μm filtered seawater and slowly warmed overnight (Gazeau et al, 2021a)."

L116, both top-down and bottom-up contribute the bacterial mortality. Shouldn't nutrient-depleted in late incubations play more important roles?

*DIP depletion may have affected growth rates at TYR and ION at the end of the experiment. This was clarified in the manuscript as* "Towards the end of the experiment bacterial growth and mortality may also have been linked to DIP depletion at TYR and ION."

L191-196, please give more details about the result instead of general description (e.g., the highest growth rates, a similar trend)

*Some details were added as follows:*

'Significant increases in heterotrophic bacterial cell specific growth rates ($p \leq 0.016$ after 24 h and 72 h) were observed in all experiments with dust under D and G (Fig. 1) relative to C, the highest growth rates relative to C were observed already 24 h after dust seeding (up to 2.7 $d^{-1}$ in G2 at FAST). Bacterial net growth rates were also higher in D and especially in G relative to C (Table 2). *Synechococcus* and picoeukaryotes net growth rates showed were also higher in D and G relative to C (Table 2). Already after 24h, in both D and G, heterotrophic bacterial mortality rates were higher than in C, especially at TYR in D (up 0.4 $d^{-1}$) and in G at ION (up to 0.3 $d^{-1}$) and FAST (up to 0.5 $d^{-1}$) (Fig. 1).'

L323-324, top-down control on the bacterioplankton would be strengthen under future conditions in the Mediterranean Sea? Too speculated.

*This statement was rephrased:*

'…..and potential increase under future conditions as suggested by the higher top-down index in G (G = 0.92 vs. C/D= 0.80, Morán et al., 2017) should be further assessed.'

L345-351, shorten this contents and added related citations.

*We revised this section according to reviewer's suggestions to:*
*"Viruses represent pivotal components of the marine food web, influencing genome evolution, community dynamics, and ecosystem biogeochemistry (Suttle, 2007). The impacts of marine viruses differ depending on whether they establish lytic or lysogenic infections (Zimmerman et al. 2019, Howard-Varona et al. 2017). Understanding how viral infection processes are influenced by changes in environmental conditions, is thus crucial to better constrain microbial mortality and cascading effects on marine ecosystems."*

Added references:

Zimmerman, A.E., Howard-Varona, C., Needham, D.M. *et al.* Metabolic and biogeochemical consequences of viral infection in aquatic ecosystems. *Nat Rev Microbiol* **18,** 21–34 (2020).

Howard-Varona, C., Hargreaves, K., Abedon, S. *et al.* Lysogeny in nature: mechanisms, impact and ecology of temperate phages. *ISME J* **11,** 1511–1520 (2017).

L481-482, Erythrobacter sp. and OM60 group are potential AAP, the authors may test pufM gene or bacteriochlorophyll using fluorescent microscopy to back this description.

*We thank the reviewer for the suggestion, we did measure bacteriochlorophyll through HPLC and have added these results as figure S9. It shows an increase in bacteriochlorophyll with dust addition in particular under future conditions. These results were added in the discussion:*
*"Moreover, bacteriochlorophyll a, a light harvesting pigment present in AAP, was generally higher in dust addition treatments especially under future conditions compared to controls (Fig. S9)."*

[Figure]

**Figure S9**: Bacteriochlorophyll a concentration measured by HPLC (see Gazeau et al 2020 for pigments measurements) over the course of the three experiments (TYR, ION and FAST).